

# The quantum entropy cone of hypergraphs

**Ning Bao[1,2]★, Newton Cheng[2]†, Sergio Hernández-Cuenca[3]‡ and Vincent P. Su[2]°**

**1** Computational Science Initiative, Brookhaven National Lab, Upton, NY, 11973, USA
**2** Center for Theoretical Physics, Department of Physics,
University of California, Berkeley, CA 94720, USA
**3** Department of Physics, University of California, Santa Barbara, CA 93106, USA

★ ningbao75@gmail.com, † newtoncheng@berkeley.edu,
‡ sergiohc@physics.ucsb.edu, ° vipasu@berkeley.edu

## Abstract

In this work, we generalize the graph-theoretic techniques used for the holographic entropy cone to study hypergraphs and their analogously-defined entropy cone. This allows us to develop a framework to efficiently compute entropies and prove inequalities satisfied by hypergraphs. In doing so, we discover a class of quantum entropy vectors which reach beyond those of holographic states and obey constraints intimately related to the ones obeyed by stabilizer states and linear ranks. We show that, at least up to 4 parties, the hypergraph cone is identical to the stabilizer entropy cone, thus demonstrating that the hypergraph framework is broadly applicable to the study of entanglement entropy. We conjecture that this equality continues to hold for higher party numbers and report on partial progress on this direction. To physically motivate this conjectured equivalence, we also propose a plausible method inspired by tensor networks to construct a quantum state from a given hypergraph such that their entropy vectors match.


# 1   Introduction

The study of entanglement entropy in holography, as originated by the Ryu-Takayanagi (RT) formula [1] and its covariant generalization [2], has had a profound impact on the conceptualization and formulation of the holographic duality, from energy conditions [3–6] and $c$-theorems [7–9] to the emergence of spacetime itself [10–13]. In addition, connections to random stabilizer states and bit threads have led to progress in better understanding the form of holographic states and their entanglement structure [14–17]. All of these ideas continue to be quite promising research directions.

A natural question to ask regarding holographic entanglement entropy is what constraints it obeys. Because of its geometric nature, it satisfies stronger entropy inequalities than those of generic quantum states. The first of these inequalities to be discovered was the monogamy of mutual information (MMI),

$$I_3(A:B:C) := S(A) + S(B) + S(C) - S(AB) - S(AC) - S(BC) + S(ABC) \leq 0, \qquad (1)$$

originally proven on time-reflection symmetric Cauchy slices [18], and later covariantly [19]. By utilizing graph-theoretic techniques, several more inequalities were discovered in [20] and used to define the holographic entropy cone of all entropy vectors allowed for holographic states with a classical geometric dual. At present, the complete set of holographic entropy inequalities obeyed by time-reflection symmetric states is known for up to and including 5 parties [20, 21], and while a general time-dependent proof remains an open question, all of them have been shown to be obeyed covariantly in specific situations [22, 23].

Recently, there has also been much work in attempting to characterize holographic entanglement and streamline the discovery of entropy inequalities by endowing entropy space with additional structures and studying useful reparameterizations thereof [24–28]. Progress along

these lines holds the potential to improve our understanding of the meaning of holographic constraints and aid in the computational tractability of finding and proving new holographic entanglement entropy inequalities.

The original motivation of the holographic entropy cone work, however, was only partially accomplished in the form of new constraints for holographic states. It was unsuccessful in finding new inequalities which could have been true for all quantum states, as in particular all holographic inequalities discovered from MMI onward are violated by Greenberger–Horne–Zeilinger (GHZ) states for sufficiently high party number [29]. The possibility to adapt holographic entanglement techniques to find results true for more general quantum states remains a tantalizing research direction, with some progress in the study of entanglement of purification [30,31], and its multipartite and conditional extensions [32–34].

In this work, we generalize the graph-theoretic techniques of [20] to the setting of hypergraphs[1] in order to capture richer, higher-partite entanglement structures. In doing so, we find a set of entropy vectors which contains the holographic entropy vectors as a strict subset. We show that the contraction map proof method for holographic entropy inequalities generalizes appropriately to hypergraphs, and use this generalization to prove new inequalities different from those obeyed by holographic states. We establish that the resulting *hypergraph entropy cone* is the same for up to 4 parties as the stabilizer entropy cone defined in [37], which is itself also equivalent to the cone defined by all balanced linear-rank inequalities [37,38]. Additionally, we make significant progress in extending these equivalences to higher party number, which we formulate in conjectural form.

The organization of the paper is as follows. In section 2, we review known results and techniques from the holographic entropy cone that are relevant for this work, and briefly review certain inequalities from the context of stabilizer states and linear ranks which turn out to be pertinent to hypergraphs. We introduce hypergraphs in section 3, with an emphasis on the features that we subsequently exploit. In the first part of this section, we prove requisite extensions of the holographic techniques from graphs to hypergraphs, and use these generalizations to prove some of the inequalities mentioned in the preceding section. In the second part, we construct extreme ray realizations in the form of hypergraphs to demonstrate completeness of the hypergraph cone for 4 parties and thereby show that it is identical to the 4-party stabilizer cone, and detail progress for 5 parties. In section 4, we propose a plausible prescription for constructing quantum states from a given hypergraph that exactly reproduce the hypergraph entropies as subsystem entropies. Finally, we conclude in section 5 with some conjectures about the equivalences of the hypergraph cone to both the stabilizer cone and the cone of balanced linear rank inequalities, as well as a discussion of potential future directions.

## 2 Review of Entropy Inequalities

In this section, we review the tools and methods used to derive entropy inequalities that apply to holographic states with a classical bulk dual. Subsequently, we briefly review several classes of entanglement entropy inequalities that are true for known subclasses of quantum states, with an eye towards assessing their validity for hypergraph entropies. Those familiar with the holographic entropy cone, stabilizer states, and linear rank inequalities may skim this section for terminology or jump to topics which are less familiar.

---

[1]It bears mentioning at this point that the graph and hypergraph states described here are not the same as the graph and hypergraph states discussed in the context of, for example, [35,36]. The construction methodology and connection to entanglement entropies differs greatly between the two formulations.

## 2.1 The Holographic Entropy Cone

Holographic states are a special subclass of quantum states whose bulk duals are classical geometries. For such states, the entanglement entropy $S(A)$ of a boundary subregion $A$ is given by the RT formula

$$S(A) = \frac{\text{Area}(\mathcal{A})}{4G}, \tag{2}$$

where $\mathcal{A}$ is the minimal-area bulk surface homologous to $A$ and anchored to its boundary, i.e. $\partial \mathcal{A} = \partial A$. The success of the holographic entropy cone methods lies in the ability to convert this holographic principle to simple combinatorial and graph constructions.

Holographic states admit simple descriptions of their entropies in terms of graphs, where entropies of subsystems are given by minimal cuts that separate their representative vertices from those of the complement. Notably, not all quantum states admit such a description. In this discrete language, we can then import the machinery of graph theory to prove entropy inequalities based on the minimality of cuts. Later, we ask which entropic constraints remain valid if one promotes graphs to hypergraphs, and find that we are able to probe non-holographic entanglement structures while preserving consistency with universal quantum inequalities.

In this section, we review the basic tools used for constructing the holographic entropy cone as applied in [20, 21], and which will be useful in our subsequent analysis of the hypergraph entropy cone. Note that some of the following definitions and results concern standard graphs only; thus, when discussing hypergraphs in section 3, only suitable generalizations of these will continue to hold.

### 2.1.1 Graph Models

All graphs considered in this paper are *undirected*. An *undirected graph* is a pair $(V, E)$ consisting of a set of vertices $V$ and a set of edges $E$ which connect pairs of vertices. Formally, edges are cardinality-2 subsets of $V$, and thus the edge set $E$ is easily seen to be a subset $E \subseteq \wp(V)$, where $\wp(V)$ denotes the power set of $V$. The *degree* of a vertex $v \in V$ is the number edges in the graph that contain $v$. To each edge, we can associate a numerical *weight* via a function $w : E \to \mathbb{R}_{\geq 0}$. The weight $|F|$ of a subset of edges $F \subseteq E$ is defined to be the sum of the individual weights: $|F| = \sum_{f \in F} w(f)$.

A *cut* is a bipartition of the vertex set $V$ into a set $W \subseteq V$ and its complement $W^{\complement} \subseteq V$, and can thus be uniquely specified by $W$ alone. Pictorially, one thinks of a cut as splitting the graph by means of *cutting* all edges bridging between $W$ and its complement. With this intuition, one defines the set of cut edges of $W$ to be

$$C(W) = \{(v, v') \in E : v \in W, v' \in W^{\complement}\}. \tag{3}$$

The *cut weight* $|C(W)|$ is defined as the sum of the weights of the cut edges. We say that a subset of vertices $W \subseteq V$ is *colored* by a set $X$ given a map $b : W \to X$, and refer to $b$ as a *coloring*. A standard choice of coloring will be the set $[n+1] := \{1, \ldots, n+1\}$. Because the motivation behind these graph models is to encode the entanglement structure of quantum states, we introduce the following nomenclature and definition of a *discrete entropy* via the min-cut prescription (see Figure 1):

**Definition 2.1.** *Let $(V, E)$ be an undirected graph with non-negative edge weights. One defines a subset of the vertex set $\partial V \subseteq V$ to be **boundary (external) vertices,** and assigns to them a coloring via a map $b : \partial V \to [n+1]$, where $n \in \mathbb{Z}^+$ is interpreted as the number of subsystems of interest in a pure quantum state on $n+1$ parties. The remaining vertices are called **bulk (internal) vertices**. Given a subset $I \subseteq [n]$, the **discrete entropy** of $I$ is defined by the cut of*

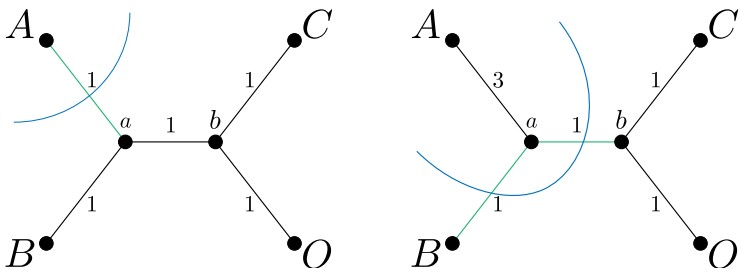

Figure 1: A simple graph with boundary vertices $A, B, C, O$ and 2 bulk vertices $a, b$. The entropy of subsystem $A$ is given by the minimal cut that separates $A$ from $B, C, O$. On the left, all edge weights are equal and the min-cut is simply given by the partition of the vertex set into $\{A\}$ and its complement. The cut is graphically represented by the blue arc, and equivalently specified by the cut edges, colored green. With different edge weights as in the right figure, the min-cut becomes $\{A, a\}$.

*minimum total weight*, or **min-cut**, *that contains precisely those boundary vertices colored by I, i.e.*

$$S(I) := \min \left\{ |C(W)| : W \cap \partial V = b^{-1}(I) \right\}. \tag{4}$$

We will often use subsets $I \subseteq [n]$ to refer to their unique pre-image $b^{-1}(I)$ in $\partial V$. In reference to vertices, we will use the terminology of *boundary* and *external* interchangeably, and similarly for *bulk* and *internal*. The quantum mechanical interpretation of the graph is accomplished by identifying the $n + 1$ boundary vertices as the avatars of the subsystems of a pure quantum state on $n + 1$ parties. To account for general mixed states, we take $n$ boundary vertices as the subsystems of interest and relegate the remaining one, usually labeled by $O$, to stand for the purification of the state. Bulk vertices do not correspond to physical subsystems; instead, they encode the entanglement structure of the quantum state. As was shown in [20], any holographic geometry can be turned into a graph whose discrete entropies match the boundary von Neumann entropies computed via the RT prescription and, conversely, graphs of the form described above can be realized geometrically with matching entropy vectors. Therefore, entanglement entropies of holographic states may be equivalently studied by considering graph models with the appropriate discrete entropies.

A general linear entropy inequality may be written

$$\sum_{l=1}^{L} \alpha_l S(I_l) \geq \sum_{r=1}^{R} \beta_r S(J_r), \tag{5}$$

where $L, R$ denote the number of terms on the LHS and RHS respectively, $I_l, J_r$ are the subsystems appearing in the inequality with non-zero coefficient, and $\alpha_l, \beta_r$ are positive coefficients. For concreteness, one might have in mind e.g. strong subadditivity (SSA),

$$S(AB) + S(BC) \geq S(ABC) + S(B) \tag{6}$$

for which $L = R = 2$ and non-trivial coefficients have $\alpha_l = \beta_r = 1$. To study this inequality one may consider graphs with no more than $n + 1 = 4$ boundary vertices $\{A, B, C, O\}$. To set the stage for future constructs, we define *n occurrence* vectors for the LHS and RHS of an inequality as

$$x_l^{(i)} := \mathbb{1}[i \in I_l], \tag{7}$$

$$y_r^{(i)} := \mathbb{1}[i \in J_r], \tag{8}$$

where $i \in [n+1]$ refers to each of the boundary vertices and $\mathbb{1}[i \in I_l]$ is the indicator function, equaling one when $i \in I_l$ and zero otherwise. Each of the $n+1$ vectors $x^{(i)}$ ($y^{(i)}$) is a bit string of length $L$ ($R$). For example, in the case of SSA, one has the following bit strings as the occurrence vector for the LHS:

$$x^A = (1,0),\tag{9}$$
$$x^B = (1,1),\tag{10}$$
$$x^C = (0,1),\tag{11}$$
$$x^O = (0,0).\tag{12}$$

### 2.1.2   The Contraction Map Method

We now review the method of *proof by contraction* developed in [20] as a tool to prove inequalities obeyed by discrete entropies on graphs. The derivation of this result is also outlined as it will be relevant to our hypergraph generalization. We first introduce the notion of a *weighted Hamming norm* $\|\cdot\|_\alpha$ on the space of $m$-dimensional bit strings $x \in \{0,1\}^m$. For non-negative weights $\alpha_l$ collected into a vector $\alpha \in \mathbb{R}^m_{\geq 0}$, this is defined as

$$\|x\|_\alpha = \sum_{l=1}^{m} \alpha_l x_l.\tag{13}$$

The contraction map method can then be stated as follows:

**Theorem 2.1.** *Let $f : \{0,1\}^L \to \{0,1\}^R$ be a $\|\cdot\|_\alpha$-$\|\cdot\|_\beta$ contraction, i.e.*

$$\left\|x - x'\right\|_\alpha \geq \left\|f(x) - f(x')\right\|_\beta \qquad \forall x, x' \in \{0,1\}^L.\tag{14}$$

*If $f\left(x^{(i)}\right) = y^{(i)}$ for all $i = 1,\dots,n+1$, then (5) is a valid entropy inequality on graphs.*

Suppose we have computed the minimal cuts for each of the subsystems $I_l$. The proof of Theorem (2.1) boils down to cutting and pasting intersections of these minimal cuts and subsequently arguing about minimality with respect to the new configuration. Bit strings are a bookkeeping tool to encode all possible such intersections of min-cuts. Note that the domain of the contraction map includes all bit strings. This is because an exhaustive contraction map $f$ must be true for all combinatorial possibilities coming from inclusion/exclusion of vertices as specified by bit strings. When searching for $f$, the only explicit constraint is the image of the $n+1$ occurrence vectors. The rest are left implicit by recursively considering the image of nearby bit strings.

*Proof.* Let $W_l$ be the minimal cut that gives the discrete entropy of the subsystem $I_l$, i.e. $S(I_l) = |C(W_l)|$. The cut associated with a bit string $x$ is defined by $W(x) = \bigcap_l W_l^{x_l}$, where the notion of inclusion/exclusion arises from declaring that $W_l^1 := W_l$ and $W_l^0 := W_l^{\complement}$. In other words, $W(x)$ is the intersection of minimal cuts $W_l$ for subsystems included by $x$ and their complements $W_l^{\complement}$ when excluded. We can piece together the minimal cut for the $I_l$ subsystem on the LHS by the following expression for bit strings $x \in \{0,1\}^L$,

$$W_l = \bigcup_{x : x_l = 1} W(x).\tag{15}$$

We now associate a cut $U_r$ to each subsystem $J_r$ that appears on the RHS of (5) by using the contraction map $f$ to pick which cuts $W(x)$ to include in the definition of $U_r$:

$$U_r = \bigcup_{x : f(x)_r = 1} W(x).\tag{16}$$

By construction, $f$ maps occurrence vectors $f\left(x^{(i)}\right) = y^{(i)}$, and thus $U_r$ provides a cut for $J_r$. The crux of this is the inclusion of specific $W(x^{(i)})$ cuts in the union above. For a given occurrence vector, the cut $W(x^{(i)})$ is essentially the minimal set of vertices that includes only the $i^{\text{th}}$ boundary vertex. For $U_r$ to be a valid cut for subsystem $J_r$, it must contain all boundary vertices that make up $J_r$, and only those. The condition that $f(x^{(i)})_r = 1$ on the occurrence vectors when building up $U_r$ means that we are taking a union of minimal cuts for these parties, thereby ensuring that every boundary vertex belonging to $J_r$ is in $U_r$.

To relate these cuts to the entropy inequalities, we expand the sum of entropies in terms of these cuts. It will be useful to introduce the notation $E(x, x') \subseteq E$ to denote the subset of the original edge set that connect vertices in $W(x)$ and $W(x')$. The edges in $E(x, x')$ cross $W_l$ if and only if $x$ and $x'$ differ in the $l^{\text{th}}$ bit. This allows us to decompose $C(W_l) = \bigcup_{x,x':x_l \neq x'_l} E(x, x')$, implying the key relation

$$|C(W_l)| = \sum_{x,x':x_l \neq x'_l} \left|E(x, x')\right|. \tag{17}$$

This can be used to rewrite the LHS of (5) as

$$\sum_{l=1}^{L} \alpha_l S(I_l) = \sum_{l=1}^{L} \alpha_l |C(W_l)| \tag{18}$$

$$= \sum_{l=1}^{L} \alpha_l \sum_{x,x':x_l \neq x'_l} \left|E(x, x')\right| \tag{19}$$

$$= \sum_{l=1}^{L} \alpha_l \sum_{x,x'} \left|x_l - x'_l\right| \left|E(x, x')\right| \tag{20}$$

$$= \sum_{x,x'} \left|E(x, x')\right| \sum_{l=1}^{L} \alpha_l \left|x_l - x'_l\right| \tag{21}$$

$$= \sum_{x,x'} \left|E(x, x')\right| \left\|x - x'\right\|_{\alpha}. \tag{22}$$

Similarly, for the $U_r$ cuts one has

$$\sum_{r=1}^{R} \beta_r |C(U_r)| = \sum_{x,x'} \left|E(x, x')\right| \left\|f(x) - f(x')\right\|_{\beta}. \tag{23}$$

The contraction property of $f$ then implies that $\sum_{r=1}^{R} \beta_r |C(U_r)| \leq \sum_{l=1}^{L} \alpha_l |C(W_l)|$.

Finally, the $U_r$ were cuts for each of the subsystems $J_r$ on the RHS of (5), but the weighted sum of their weights is lower bounded by the discrete entropies $S(J_r)$ by minimality. Putting it all together, we conclude that

$$\sum_{l=1}^{L} \alpha_l S(I_l) = \sum_{l=1}^{L} \alpha_l |C(W_l)| \geq \sum_{r=1}^{R} \beta_r |C(U_r)| \geq \sum_{r=1}^{R} \beta_r S(J_r), \tag{24}$$

and hence (5) is a valid entropy inequality. □

Note that we do not currently know if the converse of Theorem 2.1 is true; that is, an entropy inequality being valid may not imply the existence of a contraction map. We will say that an inequality *contracts* on a graph if there exists a valid contraction map that proves it.

### 2.1.3 Extreme Rays

Let us briefly review the geometric structure of the holographic entropy cone. An *entropy vector* for a state on $n$ parties is a vector in $\mathbb{R}^{2^n-1}$ *entropy space* given by calculating the von Neumann entropy of each subsystem associated to a non-empty subset of $[n]$. The set of all such entropy vectors for holographic states forms the holographic entropy cone. This cone is *convex*, i.e. it is closed under conical combinations, which are linear combinations with non-negative coefficients. The holographic entropy cone is also *polyhedral*, which means that there exists a finite set of vectors such that any other vector in it can be obtained as a conical combination of the former. The minimal set of such vectors defines the *extreme rays* of the cone and provide a complete characterization of it. A dual description of such a polyhedral cone is attained by specifying its *facets*, which are the support hyperplanes that tightly bound it. Each such facet specifies a half-space and may thus be written as a linear inequality. Every facet is a codimension-1 span of a set of extreme rays, and similarly every extreme ray is a 1-dimensional intersection of facets[2].

In order to obtain a complete characterization of an entropy cone for a given class of quantum states, it is necessary to show that every point in that entropy cone can be realized in that class. For holographic states, we are aided by the bulk/boundary duality: one does not need to scan over all possible CFT states with geometric duals, but rather simply consider all possible bulk geometries on which entropies may be computed using the RT prescription[3]. An algorithmic method to translate the geometric data necessary to calculate entanglement entropies via RT into a graph form and vice versa was first described in [20]. Thus the question of whether a given entropy vector is realizable holographically translates into a combinatorial problem of finding a graph reproducing those entropies consistently via the min-cut prescription. We note that this min-cut is a source-sink version of the problem, where we are separating boundary vertices $b^{-1}(I)$ from their complement in $\partial V$.

In applying this technique, it was shown that all extreme rays of the 3 and 4 party cones bounded by subadditivity (SA), SSA and MMI can be realized by graphs and thus by holographic states. For 5 parties, five additional inequalities were found [20] and proven to be a complete description of the holographic cone by explicit construction of graph realizations of the corresponding extreme rays [21]. Some examples of graphs corresponding to extreme rays for various numbers of parties are shown in Figure 2.

## 2.2 The Ingleton Inequality and Stabilizer States

The Ingleton inequality is a 4-party inequality of the form [40]

$$I(A:B|C) + I(A:B|D) + I(C:D) - I(A:B) \geq 0, \tag{25}$$

where $I(A:B|C) = S(AC) + S(BC) - S(ABC) - S(C)$ is the conditional mutual information of $A$ and $B$ conditioned on $C$. Importantly, up to the usual permutation and purification symmetries of the von Neumann entropy, the Ingleton inequality together with SA and SSA constitute the set of all facets that bound the 4-party entropy cone of *stabilizer states* [37]. Briefly, stabilizer states are states that can be created from the all-zeroes state with access only to phase, CNOT, and Hadamard gates, in addition to measurements. These states are ubiquitous in the field of quantum error correction [41]. The name is derived from the fact that any such state can be equivalently described as the unique common vector stabilized (i.e. eigenvector of unit eigenvalue) by some given set of operators built out of tensor products of Pauli matrices.

---

[2]Note that the converse is not true in either case. Not every codimension-1 span of extreme rays is a facet, nor is every 1-dimensional intersection of facets an extreme ray.

[3]We remain agnostic as to whether these geometries are actually dominant saddles of natural gravity path integrals. See [39] for more details on this issue.

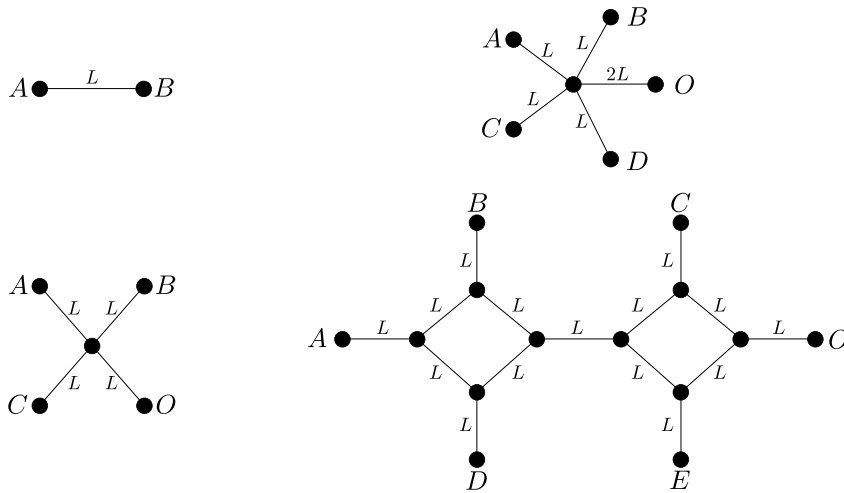

Figure 2: Graphs corresponding to extreme rays of the holographic entropy cone for various numbers of parties. The entropies of these graphs may be realized geometrically via multiboundary wormholes, where each edge corresponds to a wormhole throat whose radius is given by the edge weight. The bottom-right figure features a graph whose topology is non-trivial. Adapted from [20].

The strength of such a description is that specifying the operators is often much simpler than specifying the state itself. As an example, consider the usual single-qubit Pauli operators. Then the 3-party GHZ state,

$$|\text{GHZ}_3\rangle = \frac{1}{\sqrt{2}}(|000\rangle + |111\rangle), \tag{26}$$

is the unique state vector stabilized by the following set of operators acting on the Hilbert space of 3 qubits:

$$I_1 \otimes Z_2 \otimes Z_3, \quad Z_1 \otimes Z_2 \otimes I_3, \quad X_1 \otimes X_2 \otimes X_3. \tag{27}$$

We refer the interested reader to [42] for a more detailed account of this class of states.

For holographic states, the Ingleton inequality is implied by MMI (1), thus giving strict containment of the 4-party holographic entropy cone within the 4-party stabilizer entropy cone. These facts together make satisfaction or violation of the Ingleton inequality for subclasses of quantum states an important distinguishing feature: if a state satisfies the Ingleton inequality, it must lie within the 4-party stabilizer cone, and if it violates the Ingleton inequality, it must lie outside the stabilizer cone for any party number[4].

## 2.3 Linear Rank Inequalities

The ranks of linear subspaces of vector spaces obey certain relations known as linear rank inequalities. When applied to ranks, the Ingleton inequality is the simplest non-trivial example of these, and the only one for 4 parties (cf. 4 vector subspaces) apart from the classic Shannon inequalities [43]. All linear rank inequalities for 5 parties were found in [44] via a method based on properties of random variables with common information (in this case, vector subspaces with non-empty intersection) and partial progress has been made on more parties [45].

---

[4]Note that satisfying the Ingleton inequality is a sufficient condition for a quantum entropy vector to belong to the stabilizer entropy cone only for 4 parties. Additional inequalities need to be obeyed for more parties.

For a general number of parties, this set of inequalities naturally motivates a description of a convex cone which will be called the classical linear rank (CLR) cone.

Our interest in linear rank inequalities comes from their relation to stabilizer states. In particular, stabilizer states have been shown to obey all balanced linear rank inequalities obtained from common information [37, 38]. For up to 5 parties, this includes all linear rank inequalities except for classical monotonicity. Quantum mechanically, we will thus be interested in having SA and replacing monotonicity with weak monotonicity in the facet description of a cone analogous to the CLR cone. By performing this replacement and further completing all facet orbits according to the symmetries of the von Neumann entropy, we define the quantum linear rank (QLR) cone.

For up to and including 4 parties, the QLR and stabilizer cones are equivalent, but the extension of this equivalence to higher party number is not known due to the difficulty of defining the stabilizer entropy cone. As explained above, it is known that the stabilizer cone must be contained in the QLR cone for 5 parties, but whether additional inequalities are obeyed by stabilizer entropies remains an open question.

## 3  The Hypergraph Entropy Cone

In this section, we first introduce the basics of hypergraphs and then proceed to extend the contraction map method to hypergraphs. This allows us to show that hypergraph entropy vectors lie inside a convex, polyhedral cone that we coin the hypergraph entropy cone. This cone turns out to be bounded by inequalities that are familiar from the independently well-studied contexts of stabilizer states and linear rank inequalities. A natural question that arises is whether these valid inequalities are tight, i.e. whether they are facets of the hypergraph entropy cone. We are able to prove tightness for small numbers of parties by explicitly constructing hypergraph realizations of the extreme rays that correspond to the dual polyhedral description of the cones defined by those valid inequalities. We believe this remarkable agreement is not accidental and investigate the connection between hypergraphs and these other constructs of linear ranks and stabilizers both here and in the next section.

### 3.1  Definitions

A hypergraph $(V, E)$ is a generalization of a standard graph in which edges are promoted to hyperedges. A hyperedge is a subset of $k \geq 2$ vertices and therefore the edge set $E$ is now a more general subset $E \subseteq \wp(V)$ with no cardinality restriction on its elements other than containing at least two vertices[5]. We will refer to a hyperedge of cardinality $k$ as a $k$-edge. Let us define the *rank* of a hypergraph as the largest cardinality of hyperedges of non-zero weight in the hypergraph. A hypergraph of rank $k$ will be called a $k$-graph. We will assume that all hypergraphs in the following discussion are of finite rank. In this language, standard graphs and edges are 2-graphs and 2-edges, respectively. A hypergraph in which all hyperedges are of the same cardinality $k$ is said to be $k$-uniform.

Since hyperedges connect more than two vertices, we must revisit our rule for defining when these are cut. For a general hyperedge $e \in E$ and a given cut $W$, we include $e$ in $C(W)$ if any two of its vertices are on opposite sides of the cut. This is captured by the following definition (cf. analogous equation (3) for 2-graphs)

$$C(W) = \{e \in E : e \cap W \neq \emptyset, e \cap W^{\complement} \neq \emptyset\}. \tag{28}$$

---

[5]Single-vertex edges do not contribute to cut weights and are thus irrelevant in the current context.

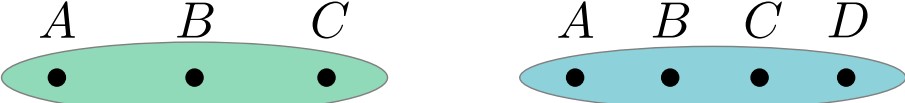

Figure 3: On the left, the entropies of the state $|\text{GHZ}_3\rangle$ are given by a hypergraph that has 3 vertices and a single 3-edge shared among them. Since entropies are computed by minimal cuts, when considering any proper, non-empty subset of vertices, the 3-edge will contribute. Hence, notably, all such subsystems have the same entropy. Similar properties hold for the state $|\text{GHZ}_4\rangle$ on the right. The entropy vector of $|\text{GHZ}_4\rangle$ is not realizable with 2-graphs. Indeed, one can easily verify that the 3-party GHZ state, corresponding to a 3-edge, satisfies MMI, while 4-party and higher GHZ states violate it.

To put it another way, $e$ does not contribute if and only if all of its vertices belong to either $W$ or $W^{\complement}$ alone. The discrete entropy will continue to be given by the total weight of the minimal cut. Just as in the 2-graph case, a hypergraph with $n + 1$ boundary vertices will be thought of as corresponding to an $n$-party quantum state[6]. The simplest hypergraph of physical interest is one that describes the entropy of the 3-party GHZ state, which consists of a single 3-edge encompassing all three vertices, as in Figure 3. Technically, the entropy vector of $|\text{GHZ}_3\rangle$ is realizable by a 2-graph without hyperedges, so one may consider $|\text{GHZ}_4\rangle$ the simplest state that requires hypergraphs. Note that the typical graphical notation of lines connecting two vertices is not suitable for hyperedges. Here, we will visualize hyperedges as subsets with different colors.

## 3.2 Contraction Map Generalization

The proof-by-contraction method can be generalized by noting that it is no longer true that the cut weight is given by a sum over edge weights $|E(x, x')|$ as in equation (17). We now need to include the contribution from higher $k$-edges, which will show up in (28). To do so, let us first define an indicator function $i^k$ on $k \geq 2$ binary bits $b_1, \ldots, b_k$ of the form

$$i^k(b_1, \ldots, b_k) = \begin{cases} 0 & \text{if} \quad b_1 = \cdots = b_k, \\ 1 & \text{otherwise.} \end{cases} \tag{29}$$

This can now be used to generalized the weighted Hamming norm introduced in (13) from 2-edges to $k$-edges. Consider $k$-many $m$-dimensional bit strings $x^1, \ldots, x^k$, such that each bit string $x^i \in \{0, 1\}^m$ consists of $m$ binary bits $x^i_1, \ldots, x^i_m$ (note the use of upper and lower indices). Then the weighted indicator function is defined as

$$i^k_\alpha(x^1, \ldots, x^k) = \sum_{l=1}^{m} \alpha_l i^k(x^1_l, \ldots, x^k_l). \tag{30}$$

The RHS is basically a sum over indicator functions, each of them acting on bits across all bit strings. More explicitly, stacking the $k$ bit strings of length $m$ as the rows of a $k \times m$ matrix,

---

[6]Our usage of the word "state" suggests that hypergraphs can indeed represent physical quantum states – we take this as an assumption for now, and provide some motivation for it in later sections. Evidence supporting that this prescription is still meaningful quantum mechanically will be 2-fold: for small party numbers, 1) we show that the resulting entropy vectors lie strictly inside the quantum entropy cone (see section 3.3) and 2) we are able to propose a plausible prescription to construct a quantum state given a hypergraph such that their entropies match exactly (see section 4).

one may think of the $l^{\text{th}}$ term in the sum as corresponding to the evaluation of the indicator function on the $k$ bits in the $l^{\text{th}}$ column. It is easy to see that $i_\alpha^2(x, x') = \left\| x - x' \right\|_\alpha$, thus generalizing the weighted Hamming norm in (13) in a consistent way.

Define now $E(x^1, \ldots, x^k)$ as the set of all $k$-edges across all cuts in $W(x^1), \ldots, W(x^k)$[7], where each of the latter are defined via inclusion/exclusion as in the proof of (2.1). For a cut $W_l$ of some subsystem $I_l$, the contribution to $|C(W_l)|$ coming from $k$-edges is

$$|C_k(W_l)| = \sum_{(x^1, \ldots, x^k): x_l^i \neq x_l^j} \left| E(x^1, \ldots, x^k) \right|, \tag{31}$$

where the sum is over the set of all $k$-tuples of bit strings such that any two $x^i, x^j$ of them differ in their $l^{\text{th}}$ bit (which is the necessary and sufficient condition for a $k$-edge to be cut by $W_l$). The total cut weight is then just the sum of the contributions from $k$-edges of every possible cardinality $k$:

$$|C(W_l)| = \sum_k |C_k(W_l)| = \sum_{(x^1, x^2): x_l^i \neq x_l^j} \left| E(x^1, x^2) \right| + \sum_{(x^1, x^2, x^3): x_l^i \neq x_l^j} \left| E(x^1, x^2, x^3) \right| + \ldots. \tag{32}$$

As suggested by the decomposition above, we can break up the hard problem of studying a candidate inequality on $k$-graphs into $k - 1$ simpler problems. Namely, one can just check validity on all $t$-uniform graphs with $2 \leq t \leq k$, so that only hyperedges of cardinality $t$ appear in each case. That this is a necessary condition is clear: an inequality can only possibly hold on $k$-graphs if it is valid on all $t$-uniform hypergraphs of smaller or equal rank since the latter are just a subclass of the former[8]. That it is sufficient follows from the decomposition into fixed-rank contributions exhibited in (32): if the contribution to $|C(W_l)|$ from each possible rank $t$ respects a given inequality, then by linearity of (32) the sum of all contributions for $2 \leq t \leq k$ will obey that inequality too.

It follows from these observations that the problem of proving inequalities for hypergraphs should reduce to that of proving inequalities for $k$-uniform graphs (see Corollary 3.1.1 below for a formalization and proof of this statement). Thus consider a candidate entropy inequality of the form (5) and an arbitrary $k$-uniform graph. Then the LHS can be written as

$$\sum_{l=1}^L \alpha_l S(I_l) = \sum_{l=1}^L \alpha_l \left| C_k(W_l) \right| \tag{33}$$

$$= \sum_{l=1}^L \alpha_l \sum_{(x^1, \ldots, x^k): x_l^i \neq x_l^j} \left| E(x^1, \ldots, x^k) \right| \tag{34}$$

$$= \sum_{l=1}^L \alpha_l \sum_{(x^1, \ldots, x^k)} i^k(x_l^1, \ldots, x_l^k) \left| E(x^1, \ldots, x^k) \right| \tag{35}$$

$$= \sum_{(x^1, \ldots, x^k)} \left| E(x^1, \ldots, x^k) \right| \sum_{l=1}^L \alpha_l i^k(x_l^1, \ldots, x_l^k) \tag{36}$$

$$= \sum_{(x^1, \ldots, x^k)} \left| E(x^1, \ldots, x^k) \right| i_\alpha^k(x^1, \ldots, x^k), \tag{37}$$

---

[7]This means that each $k$-edge belongs to every one of the $W(x^i)$ cuts in the sense of (28), such that its vertices non-trivially intersect all of the $W(x^i)$.

[8]In passing we note that this agrees with the expectation that the inequalities obeyed by hypergraphs should weaken as higher-cardinality edges become available. For example, an inequality that is valid for 3-graphs is automatically valid for any standard 2-graph, because the latter is just an example of the former where all 3-edges are trivial. The converse is not true: an inequality that is valid for 2-graphs will not necessarily hold on 3-graphs.

and similarly for the RHS. The remainder of the proof of the contraction theorem for 2-graphs then extrapolates identically to the case of $k$-uniform graphs. We therefore conclude that the generalization of Theorem 2.1 can be stated as follows:

**Theorem 3.1.** *Let $f : \{0,1\}^L \to \{0,1\}^R$ be an $i_\alpha^k$-$i_\beta^k$ contraction:*

$$i_\alpha^k(x^1,\ldots,x^k) \geq i_\beta^k(f(x^1),\ldots,f(x^k)) \qquad \forall x^1,\ldots,x^k \in \{0,1\}^L. \tag{38}$$

*If $f\left(x^{(i)}\right) = y^{(i)}$ for all $i \in [n+1]$, then (5) is a valid entropy inequality on $k$-uniform graphs.*

Clearly, this immediately reduces to Theorem 2.1 for $k = 2$. An important and immediate corollary is the following:

**Corollary 3.1.1.** *If (5) contracts on $k$-uniform graphs, then it contracts on $k$-graphs.*

*Proof.* By assumption, $i_\alpha^k(x^1,\ldots,x^k) \geq i_\beta^k(f(x^1),\ldots,f(x^k))$ for all choices of $k$ LHS bit strings $x^1,\ldots,x^k \in \{0,1\}^L$. This includes choices with repeated bit strings. Whenever only $l \geq 2$ out of the $k$ bit strings on the LHS are distinct, the $i_\alpha^k$-$i_\beta^k$ contraction reduces to an $i_\alpha^l$-$i_\beta^l$ contraction. Since among all subsets of $k$ LHS bit strings one has all subsets of $l$ distinct LHS bit strings for every $2 \leq l \leq k$, $f$ is a contraction on all $k$-graphs. $\qquad\square$

Note that, as mentioned previously, a valid entropy inequality on $k$-graphs will not generally hold on $k'$-graphs with $k' > k$. In the holographic case, we were only concerned with 2-edges, implying that holographic inequalities proven via the contraction method do not a priori hold on general hypergraphs. Intuitively, as $k$ increases the number of restrictions on the contraction map also increases. This means that only weaker entropy inequalities will be valid, opening up the cone to less stringent facets, which in turn implies that the hypergraph cone is guaranteed to contain the holographic cone, as expected.

Theorem 3.1 alone is not very satisfying if one hopes to make general statements about the entropies of hypergraphs of arbitrarily high rank. For example, if we want to interpret hypergraphs as truly encoding the entanglement structure of some class of quantum states, their entropies should satisfy universal entropy inequalities such as SA and SSA, obeyed by all quantum states, regardless of their rank. To check such a fundamental consistency condition, one ideally need not verify the validity of these inequalities for $k$ graphs of arbitrarily large $k$. Fortunately, the following result tightly bounds how high in rank one needs to go in order to prove that a certain inequality is valid for hypergraphs of all ranks.

**Proposition 3.1.1.** *If (5) contracts on $R$-graphs, then it is a valid inequality on all hypergraphs of finite rank.*

*Proof.* The crux of this proof is to show that for $k > R$, the $i^k$-distance between a set of $k$ bit strings of length $R$ is equal to the $i^t$ distance of a subset of at most $t \leq R$ of them. This will be shown to be a consequence of the fact that the $i^k$-distance saturates at a maximum value as $k$ increases above $R$. We then show that this reduces higher rank contraction map constraints to rank $R$ constraints.

Consider an arbitrary set of $k \geq R+1$ distinct bit strings $Z^k = \{x^i \in \{0,1\}^R : i = 1,\ldots,k\}$. Define a subset $C \subseteq [R]$ of RHS columns such that $i^k(x_l^1,\ldots,x_l^k) = 1$ if and only if $l \in C$, and note that $i_\beta^k(x^1,\ldots,x^k) = \sum_{l \in C} \beta_l$. Clearly, $C$ is non-empty for any non-trivial inequality with $R \geq 1$, and its cardinality $|C| \geq 2$ for any $k \geq 3$. The $k = 2$ case, which may only occur for $R = 1$, is trivial.

Write $C = \{l_1,\ldots,l_{|C|}\}$ for convenience and proceed to construct a subset $S \subseteq Z^k$ algorithmically as follows. First, pick two bit strings $x^{j_1}, x^{j_2} \in Z^k$ with the property that

$i^2(x_{l_1}^{j_1}, x_{l_1}^{j_2}) = i^2(x_{l_2}^{j_1}, x_{l_2}^{j_2}) = 1$ for $l_1, l_2 \in C$ and add them to $S$. These are two bit strings differing in the $l^{\text{th}}$ bit, which are guaranteed to exist so long as $k \geq 3$. Then, for the next $i_3 \in C$, look at $i^2(x_{l_3}^{j_1}, x_{l_3}^{j_2})$. If $i^2(x_{l_3}^{j_1}, x_{l_3}^{j_2}) = 0$, then look for a third bit string $x^{j_3}$ giving $i^3(x_{l_3}^{j_1}, x_{l_3}^{j_2}, x_{l_3}^{j_3}) = 1$, which exists because $l_3 \in C$, and add it to $S$. If instead $i^2(x_{l_3}^{j_1}, x_{l_3}^{j_2}) = 1$, do nothing and go on to look at the next $l_4 \in C$. The process terminates once one goes over all elements in $C$, at the end of which one ends up with a set $S$ containing $|S| \leq R$ bit strings and with the property that $i^{|S|}(x_{l_c}^{j_1}, \ldots, x_{l_c}^{j_{|S|}}) = 1$ for all $l_c \in C$. This implies that

$$i_\beta^{|S|}(x^{j_1}, \ldots, x^{j_{|S|}}) = i_\beta^k(x^1, \ldots, x^k). \tag{39}$$

By hypothesis, there exists a contraction map $f : \{0,1\}^L \rightarrow \{0,1\}^R$ such that $i_\alpha^m(\tilde{x}^1, \ldots, \tilde{x}^m) \geq i_\beta^m(f(\tilde{x}^1), \ldots, f(\tilde{x}^m))$ for all $\tilde{x}^1, \ldots, \tilde{x}^k \in \{0,1\}^L$ and all $m \leq R$. In particular, this implies $i_\alpha^{|S|}(\tilde{x}^{j_1}, \ldots, \tilde{x}^{j_{|S|}}) \geq i_\beta^{|S|}(x^{j_1}, \ldots, x^{j_{|S|}})$ where $f(\tilde{x}^{j_s}) = x^{j_s}$ for all $s = 1, \ldots, |S|$. Using on the LHS the fact that the weighted indicator functions are monotonically non-decreasing upon addition of extra points, and (39) on the RHS, one arrives at $i_\alpha^k(\tilde{x}^{j_1}, \ldots, \tilde{x}^{j_k}) \geq i_\beta^k(x^{j_1}, \ldots, x^{j_k})$. Since the set $Z^k$ was arbitrary, this shows that $f$ is an $i_\alpha^k$-$i_\beta^k$ contraction. Additionally, since $k \geq R + 1$ was also arbitrary, and $f$ is already an $i_\alpha^m$-$i_\beta^m$ contraction for all $m \leq R$ by assumption, one concludes that (39) is a valid inequality on all hypergraphs. $\square$

Proposition 3.1.1 partially captures the intuition that one should not need to consider arbitrarily large entanglement structures given a fixed party number in order to confirm the validity of an entropy inequality. This is realized as a bound on the maximal rank $k$ of $k$-graphs one has to consider of the form $k \leq R$, where $R$ is generically smaller for smaller numbers of parties. Indeed, we can slightly formalize this statement as follows. Given $n$ parties, there are $2^n - 1$ non-trivial combinations of parties or subsystems. For inequalities that only have unit coefficients, a generic inequality can have at most $2^{n-1} - 1$ terms on the RHS, or else they will be violated by sufficiently high party GHZ states[9]. Including non-unit coefficients will modify this bound by a multiplicative factor that depends on the ratio of the coefficients. We hence expect that there is generally an $O(2^{n-1} - 1)$ bound on the rank of the hypergraphs to be considered for generic $n$-party inequalities after which the contraction maps begin to trivialize.

However, on the basis of physical intuition, we expect that the bound should be stronger than one exponential in the party number. Including the purifier, a general $n$-party quantum state can only possibly accommodate at most $(n+1)$-party entanglement among the indivisible subsystems. This leads to the expectation that $k$-edges, which are $k$-partite entanglement structures of $\text{GHZ}_k$ type, should not contribute anything new to an $n$-party hypergraph with $n + 1 < k$. In particular, one would hope that any $n$-party hypergraphs should be reducible to an entropically-equivalent one of rank at most $n + 1$. As for inequalities, this would mean that we only need to check the contraction property on up to $(n+1)$-graphs when considering $n$-party inequalities. We phrase this intuition here as a formal conjecture:

**Conjecture 3.1.** *If an $n$-party inequality of the form* (5) *contracts on* $(n+1)$*-graphs, then it is a valid inequality on all hypergraphs of finite rank.*

Although we will leave further investigation of this conjecture to future work, we briefly point out a corollary of 3.1.1 which sharpens the sense in which the constraints on the contraction map coming from $k$-edges weaken as $k$ increases. The result below shows that, except for very specific choices of RHS bit strings, the contraction property is guaranteed on $k$-edges by the contraction property of $l$-edges with $l < k$.

---

[9]Recall that a $\text{GHZ}_k$ state results in all $2^k - 2$ non-trivial proper subsystems having the same entropy.

**Corollary 3.1.2.** *If $f$ is an $i^l_\alpha$-$i^l_\beta$ contraction for all $m \le k-1$, then further demanding that $i^k_\alpha(x^1, \ldots, x^k) \ge i^k_\beta(f(x^1), \ldots, f(x^k))$ is a non-trivial constraint on $f$ if and only if $i^k_\beta(f(x^1), \ldots, f(x^k)) > i^{k-1}_\beta(f(x^{i_1}), \ldots, f(x^{i_{k-1}}))$ strictly for any subset of $k-1$ bit strings of the original set of $k$ bit strings.*

*Proof.* This follows immediately from the observation in the proof of Proposition 3.1.1 that given $k$ RHS bit strings, one can always form a subset $S$ of $|S| \ge k$ bit strings obeying (39). Whenever $|S| < k$ strictly, then that $i^k_\alpha(x^1, \ldots, x^k) \ge i^k_\beta(f(x^1), \ldots, f(x^k))$ holds follows from the contraction property for $m = |S| \le k-1$. Hence such constraint is only non-trivial (in the sense of not being implied by $m \le k-1$) if $|S| = k$, which happens if and only if all $k$ bit strings are needed for the weighted indicator function to preserve its value on the RHS. In other words, one needs that $i^k_\beta(f(x^1), \ldots, f(x^k)) > i^{k-1}_\beta(f(x^{i_1}), \ldots, f(x^{i_{k-1}}))$ strictly for any subset of $k-1$ bit strings of the original set of $k$ bit strings. $\square$

### 3.2.1 Polyhedrality

The contraction map technique only works to prove linear inequalities in the form of (5). Here, we show that the hypergraph cone is polyhedral (and therefore convex), and hence one need only consider such linear inequalities to completely characterize it.

In [20], the holographic entropy cone was shown to be polyhedral for any fixed party number $n$, implying the existence of finitely many linearly independent entropy inequalities for $n$-party holographic states. Polyhedrality is a remarkable property not necessarily shared by other entropy cones, such as the Shannon entropy cone [38]. By borrowing the proof techniques for 2-graphs from [20], we can prove that the hypergraph cone is also polyhedral. The following lemma is key:

**Lemma 3.1.1.** *Any entropy vector in the n-party hypergraph entropy cone can be realized by a hypergraph with $2^{2^n-1}$ vertices.*

*Proof.* Consider $n$ parties $i \in [n]$ and let $\mathcal{I}_n$ denote the set of $2^n-1$ non-trivial combinations of parties or subsystems. We can then define a universal hypergraph with a vertex set $V = \{0, 1\}^{\mathcal{I}_n}$ of all bit strings of length $2^n - 1$, whence $|V| = 2^{2^n-1}$. The boundary vertices in this set are precisely the occurrence vectors for each $i$; that is, they correspond to those bit strings $x^i$ defined via $x^i_I = \mathbb{1}[i \in I \in \mathcal{I}_n]$.

Now given an arbitrary $n$-party hypergraph, let $W_I$ be a min-cut giving the entropy of subsystem $I$. Then for each $x \in V$, we define a cut $W(x)$ using inclusion/exclusion as in the proofs of Theorems 2.1 and 3.1: $W(x) = \bigcap_{I \in \mathcal{I}_n} W^{x_I}_I$ where $W^1_I = W_I$ and $W^0_I = W^{\complement}_I$. The set of $W(x)$ for all possible $x \in V$ partitions the hypergraph into $2^{2^n-1}$ subsets, some of which may be empty. The idea is to map each of these subsets $W(x)$ to a single vertex $x \in V$ in the universal hypergraph, which may alternatively be thought of as collapsing $W(x)$ into a single vertex in the given hypergraph. One then only needs to make sure that the entropies can be appropriately preserved. To do so, let $E(x^1, \ldots, x^k)$ denote the set of $k$-edges on the given hypergraph intersecting non-trivially every one of the cuts $W(x^1), \ldots, W(x^k)$. Note that the $x^i$ are allowed to be equal, in order to account for the possibility of a $k$-edge having multiple vertices in the same $W(x^i)$. Then define a $k$-edge containing vertices $(x^1, \ldots, x^k)$ in the universal hypergraph with weight given by the sum of all $k$-edges across $W(x^1), \ldots, W(x^k)$ so that $w(x^1, \ldots, x^k) = \sum_{e \in E(x^1, \ldots, x^k)} w(e)$. Because the cuts $W_I$ are only concerned with the hyperedges that cross the partition, it is clear that this indeed preserves the entropies. $\square$

Applying this lemma, the proof of polyhedrality of the hypergraph entropy cone is identical to the proof of Proposition 7 in [20], so long as one replaces "graph" with "hypergraph." This

proves that the hypergraph entropy cone is polyhedral and can thus be described by a finite number of facets for any given finite number of parties. Moreover, convexity of the hypergraph cone is an immediate corollary, implying that the cone can be fully described by linear inequalities.

### 3.2.2 A Geometric Aside: The Hypercube Picture

A potentially useful framing of Theorem 3.1 is to consider all possible bit strings $x$ of $m$ bits as a labeling scheme for all vertices of an $m$-dimensional hypercube such that any two bit strings differing only by the $l^{\text{th}}$ bit are connected by an edge along the $l^{\text{th}}$ dimension. Any two bit strings, adjacent or not, can be used as generators of a "straight" line connecting them. The quotes are used to emphasize that this line has to travel along the edges of the hypercube, and will thus be piece-wise broken at any turning-vertex that has to be traversed to connect the two vertices. This line follows a shortest-distance path which is highly non-unique for the Hamming distance function $\|\cdot\|$. For any two bit strings $x^1$ and $x^2$, a third bit string $x^3$ can be said to be aligned between $x^1$ and $x^2$ if it lies along any such straight line through the latter. This happens when the three bit strings saturate the triangle inequality

$$\left\|x^1 - x^2\right\| \leq \left\|x^1 - x^3\right\| + \left\|x^2 - x^3\right\|. \tag{40}$$

This is naturally consistent with the intuition that the three bit strings are aligned and that, as such, still span a 1-dimensional object. In contrast, if the triangle inequality is not saturated, then they no longer span a line, but a 2-dimensional object. Geometrically, the increase in dimensionality is due to the need to move in an additional direction, apart from (any one of) the direction(s) from $x_1$ to $x_2$, to additionally reach $x_3$. At the level of the bit string, this can be seen as a consequence of there not existing a minimal sequence of bit flips from $x_1$ to $x_2$ that additionally realizes $x_3$. One may picture the three bit strings as labeling the vertices of a 2-dimensional polytope which is non-degenerate in the sense that it has dimension $v - 1$, where $v$ is its number of vertices (cf. the three aligned vertices, which form a degenerate 1-dimensional polytope).

Suppose one adds a fourth bit string $x^4$ and asks whether the resulting 4-vertex polytope is degenerate. From the geometric perspective, it will be degenerate if $x^4$ can be reached by following any of the paths that minimize the distance one has to travel to connect the other $x^1$, $x^2$ and $x^3$ vertices alone, and one may picture the fourth vertex as lying inside the 2-dimensional polytope spanned by the other three vertices (cf. the traveling salesman problem if one adds a city along one of the already optimal paths). In the language of bit strings, the polytope is degenerate if any one of the minimal sequences of bit flips it takes to go from $x_1$ to $x_2$ and $x_3$ also realizes $x_4$ along the way with no extra cost. More explicitly, this will happen if and only if all bits in which $x^4$ differs from each of the other three bit strings are a subset of the bits in which the latter alone already collectively differ. Crucially, this is intimately related to the indicator function $i_\beta^k$, which may now pictorially be thought of as measuring the volume of a polytope specified by $k$ hypercube vertices! More precisely, one has the following result:

**Proposition 3.1.2.** *A set of $k$ bit strings $x^1, \ldots, x^k$ span a $(k-1)$-dimensional polytope if and only if $i_\beta^k(x^1, \ldots, x^k) > i_\beta^{k-1}(x^{l_1}, \ldots, x^{l_{k-1}})$ strictly for all subsets of $k-1$ bit strings.*

*Proof.* Suppose $i_\beta^k(x^1, \ldots, x^k) = i_\beta^{k-1}(x^{l_1}, \ldots, x^{l_{k-1}})$ for some subset of $k-1$ bit strings. This means that the $l_k^{\text{th}}$ bit string $x^{l_k}$ happens to contribute nothing to the $i_\beta^k$ function, such that the latter trivializes down to an $i_\beta^{k-1}$ function (cf. measuring a higher-dimensional volume of a lower-dimensional object). This happens if and only if all bits in which $x^k$ differs from each of the other $k-1$ bit strings are a subset of the bits in which the latter alone already

collectively differ. As explained above, this is precisely the situation in which the $x^{l_k}$ bit string labels a vertex inside the polytope defined by the bit strings $x^{l_1}, \ldots, x^{l_{k-1}}$, thus not raising its dimension and proving that the $x^1, \ldots, x^k$ vertices span a polytope of dimension at most $k - 2$. $\qquad\square$

With this intuition, the statement of corollary 3.1.2 essentially says that the only constraints on the contraction map from $k$-graphs that are not already provided by $(k-1)$-graphs come from choices of RHS vertices that generate full-dimensional polytopes. Because the indicator function $i_\beta^k$ effectively measures a $(k-1)$-dimensional volume, it trivializes to an $i_\beta^l$ function with $l < k$ if the object being measured is a polytope of lower dimension $l - 1$. Similarly, Proposition 3.1.1 can be rephrased as saying that in an $R$-dimensional RHS hypercube, any choice of vertices will give a polytope of dimension at most $R - 1$[10], and therefore any $i_\beta^k$ function with $k > R$ will trivialize to some $i_\beta^l$ with $l \leq R$.

This hypercube picture also appeared in [20], in which it was shown that Theorem 2.1 can equivalently be stated as requiring that a valid contraction map should never increase the graph distance between two points on the hypercube. We suspect that a similar picture naturally emerges here when interpreting the $k$-distance in Theorem 3.1 as in the discussion above, and leave it for future exploration.

### 3.2.3 Explicit Analysis of Simple Inequalities

Here we look at some basic inequalities for small party numbers and walk the reader through the logic involved in the application of the hypergraph contraction map method presented in the previous section.

We begin by noting that any hope for an entropic interpretation of hypergraphs demands that they satisfy SA, viz.

$$S(A) + S(B) \geq S(AB). \tag{41}$$

In this simplest case, one can easily see that the RHS is too small to accommodate any contribution to the distance function from $k$-edges with $k \geq 3$. Noting that SA is valid for 2-graphs, application of Proposition 3.1.1 implies that SA is indeed valid for all hypergraphs[11]. Because 2-graphs already fill up the cone defined by SA alone, it follows that SA is the only inequality on 2 parties obeyed by hypergraphs.

We now move on to 3 parties, where 2-graphs are already entropically more restricted than general quantum states because they satisfy MMI. While MMI is easily violated by hypergraphs, consistency with quantum mechanics still demands that SSA, given in (6), be a valid inequality. Applying Proposition 3.1.1, we just need SSA to be valid on 2-graphs for it to hold in general. Because SSA contains so few terms, its contraction map is completely determined by its occurrence vectors, as in Table 1. It is then simple to verify by inspection that SSA indeed holds for 2-uniform graphs.

Table 1: Tabular representation of the 2-graph contraction map for SSA.

|   | $AB$ | $BC$ | $B$ | $ABC$ |
|---|---|---|---|---|
| $A$ | 1 | 0 | 0 | 1 |
| $B$ | 1 | 1 | 1 | 1 |
| $C$ | 0 | 1 | 0 | 1 |
| $O$ | 0 | 0 | 0 | 0 |

---

[10]Note that the hypercube is "hollow".

[11]In fact, one might simply argue for the validity of SA as a trivial consequence of the minimality condition in the definition of the discrete entropy (cf. SA in the holographic context).

Perhaps unsurprisingly, neither MMI nor any of the other known holographic entanglement entropy inequalities for higher party number are obeyed by generic hypergraphs. A simple way of seeing this is to note that all of these inequalities happen to have more entropies on the RHS than on the LHS when written as in (5). Any such inequality is immediately violated by a GHZ state for a sufficiently high number of parties, corresponding to a single hyperedge of sufficiently high cardinality. As an example, MMI has 4 terms on the RHS but only 3 on the LHS, and is thus violated by GHZ states on 4 or more parties. It is instructive to see how the unique 2-graph contraction map for MMI, shown in Table 2, captures this fact by failing to obey the contraction property for 4-edges[12].

> Table 2: Tabular representation of the 2-graph contraction map for the holographic inequality MMI, given in equation 1. It is easy to see by inspection that 3-edges still obey the contraction property, so that MMI holds for 3-graphs. This is however no longer true for 4-edges, which violate the contraction property already at the level of the occurrence vectors. Indeed, these give an $i^4$ distance of 3 on the LHS, but map to an $i^4$ distance of 4 on the RHS, thus not contracting.

|   | $AB$ | $BC$ | $AC$ | $A$ | $B$ | $C$ | $ABC$ |
|---|------|------|------|-----|-----|-----|-------|
| $O$ | 0 | 0 | 0 | 0 | 0 | 0 | 0 |
|   | 0 | 0 | 1 | 0 | 0 | 0 | 1 |
|   | 0 | 1 | 0 | 0 | 0 | 0 | 1 |
| $C$ | 0 | 1 | 1 | 0 | 0 | 1 | 1 |
|   | 1 | 0 | 0 | 0 | 0 | 0 | 1 |
| $A$ | 1 | 0 | 1 | 1 | 0 | 0 | 1 |
| $B$ | 1 | 1 | 0 | 0 | 1 | 0 | 1 |
|   | 1 | 1 | 1 | 0 | 0 | 0 | 1 |

One may wonder whether, apart from SA and SSA, any other inequality that is not MMI could hold on hypergraphs for 3 parties. The answer turns out to be negative and follows from realizability of extreme rays and convexity, as usual. In particular, one notes that the extreme rays of the cone whose facets are SA and SSA correspond to the entropies of Bell pairs, 4-partite perfect tensors and the 4-partite GHZ state. Since all of these are realizable by hypergraphs[13], it follows that SA and SSA are the only facets that bound the hypergraph entropy cone for 3 parties.

The simplest entropy inequality obeyed by hypergraphs that is not quantum mechanical occurs at 4 parties and, interestingly, is the Ingleton inequality. The proof of this inequality for hypergraphs is best obtained by direct implementation of the generalized contraction map method explained in section 3.2. In Table 3, we show that the Ingleton inequality indeed holds on 5-graphs, and because $R = 5$ for the Ingleton inequality, Proposition 3.1.1 establishes that hypergraphs of arbitrary finite rank obey it as well.

## 3.3 Explicit Constructions of the Hypergraph Cone

To study the hypergraph entropy cone more systematically, we take two lines of inquiry. One approach is to ask what candidate inequalities might bound the hypergraph cone and try to prove them using the contraction map method. The other is to ask which candidate entropy vectors may be inside the hypergraph cone and try to realize them using hypergraphs by solving a particular integer linear program. Although there are many drawbacks to this strategy,

---

[12]In producing this table, note that the only constraints on the contraction map $f$ one starts with are the bit strings corresponding to boundary vertices $A, B, C, O$ and their images, which violate the contraction property of $i^4$ by themselves. The remaining images of $f$ were allowed to vary only subject to the contraction constraint, but the $i^4$ distance was already doomed to not contract. Nevertheless, the map shown does obey the contraction property for the $i^2$ and $i^3$ distances, thus proving MMI on 3-graphs.

[13]Note that Bell pairs and perfect tensors are already realizable by holographic 2-graphs.

Table 3: Tabular representation of a contraction map which proves that the Ingleton inequality is valid on hypergraphs. The right-most column $f_{10}$ is a compact decimal-base representation of the map images understood as digits of a binary number (e.g. $10011_2 = 19_{10}$ for the occurrence image of $A$). Given an arbitrary inequality, with entropies canonically ordered lexicographically and the map domain in increasing order, the list of values of $f_{10}$ is a succinct specification of the contraction map.

|   | AB | AC | AD | BC | BD | A | B | CD | ABC | ABD | $f_{10}$ |
|---|----|----|----|----|----|---|---|----|-----|-----|----------|
| O | 0 | 0 | 0 | 0 | 0 | 0 | 0 | 0 | 0 | 0 | 0 |
|   | 0 | 0 | 0 | 0 | 1 | 0 | 0 | 0 | 0 | 1 | 1 |
|   | 0 | 0 | 0 | 1 | 0 | 0 | 0 | 0 | 1 | 0 | 2 |
|   | 0 | 0 | 0 | 1 | 1 | 0 | 0 | 0 | 1 | 1 | 3 |
|   | 0 | 0 | 1 | 0 | 0 | 0 | 0 | 0 | 0 | 1 | 1 |
| D | 0 | 0 | 1 | 0 | 1 | 0 | 0 | 1 | 0 | 1 | 5 |
|   | 0 | 0 | 1 | 1 | 0 | 0 | 0 | 0 | 0 | 0 | 0 |
|   | 0 | 0 | 1 | 1 | 1 | 0 | 0 | 0 | 0 | 1 | 1 |
|   | 0 | 1 | 0 | 0 | 0 | 0 | 0 | 0 | 1 | 0 | 2 |
|   | 0 | 1 | 0 | 0 | 1 | 0 | 0 | 0 | 0 | 0 | 0 |
| C | 0 | 1 | 0 | 1 | 0 | 0 | 0 | 1 | 1 | 0 | 6 |
|   | 0 | 1 | 0 | 1 | 1 | 0 | 0 | 0 | 1 | 0 | 2 |
|   | 0 | 1 | 1 | 0 | 0 | 0 | 0 | 0 | 1 | 1 | 3 |
|   | 0 | 1 | 1 | 0 | 1 | 0 | 0 | 0 | 0 | 1 | 1 |
|   | 0 | 1 | 1 | 1 | 0 | 0 | 0 | 0 | 1 | 0 | 2 |
|   | 0 | 1 | 1 | 1 | 1 | 0 | 0 | 0 | 0 | 0 | 0 |
|   | 1 | 0 | 0 | 0 | 0 | 0 | 0 | 0 | 0 | 1 | 1 |
|   | 1 | 0 | 0 | 0 | 1 | 0 | 0 | 0 | 1 | 1 | 3 |
|   | 1 | 0 | 0 | 1 | 0 | 0 | 0 | 0 | 1 | 1 | 3 |
| B | 1 | 0 | 0 | 1 | 1 | 0 | 1 | 0 | 1 | 1 | 11 |
|   | 1 | 0 | 1 | 0 | 0 | 0 | 0 | 0 | 1 | 1 | 3 |
|   | 1 | 0 | 1 | 0 | 1 | 0 | 0 | 0 | 0 | 1 | 1 |
|   | 1 | 0 | 1 | 1 | 0 | 0 | 0 | 0 | 0 | 1 | 1 |
|   | 1 | 0 | 1 | 1 | 1 | 0 | 0 | 0 | 1 | 1 | 3 |
|   | 1 | 1 | 0 | 0 | 0 | 0 | 0 | 0 | 1 | 1 | 3 |
|   | 1 | 1 | 0 | 0 | 1 | 0 | 0 | 0 | 0 | 1 | 1 |
|   | 1 | 1 | 0 | 1 | 0 | 0 | 0 | 0 | 1 | 0 | 2 |
|   | 1 | 1 | 0 | 1 | 1 | 0 | 0 | 0 | 1 | 1 | 3 |
| A | 1 | 1 | 1 | 0 | 0 | 1 | 0 | 0 | 1 | 1 | 19 |
|   | 1 | 1 | 1 | 0 | 1 | 0 | 0 | 0 | 1 | 1 | 3 |
|   | 1 | 1 | 1 | 1 | 0 | 0 | 0 | 0 | 1 | 1 | 3 |
|   | 1 | 1 | 1 | 1 | 1 | 0 | 0 | 0 | 0 | 1 | 1 |

improvements to it are scarce and only heuristic, which is one of the reasons why constructing this and other entropy cones is hard. The most obvious limitation is that in principle we do not actually have a systematic way of generating "good" candidate inequalities and entropy vectors. In practice, however, we have found a powerful heuristic. The fact that hypergraphs obey the Ingleton inequality, the simplest of the well-studied family of linear rank inequalities, suggests a connection between hypergraphs and the QLR cone (see section 2.3). Exploiting this connection turns out to be a remarkably fruitful direction and strongly suggestive that this connection is no accident.

In what follows, we provide an explicit construction of the hypergraph entropy cone for 4 parties, which allows us to prove that it identically matches the QLR and stabilizer cones at the same party number. For 5 parties, we take as candidate inequalities the facets of the QLR cone and as candidate rays the extreme rays of the CLR cone[14]. We are able to realize all such extreme rays and to partially prove all such inequalities, which strongly hints at an agreement

---

[14]Unfortunately, the polyhedral conversion from the facet description of the QLR to its extreme ray description was out of the scope of our computational resources. We believe that, should one compute them, the extreme rays of the QLR cone will also be good candidates and likely realizable by hypergraphs too.

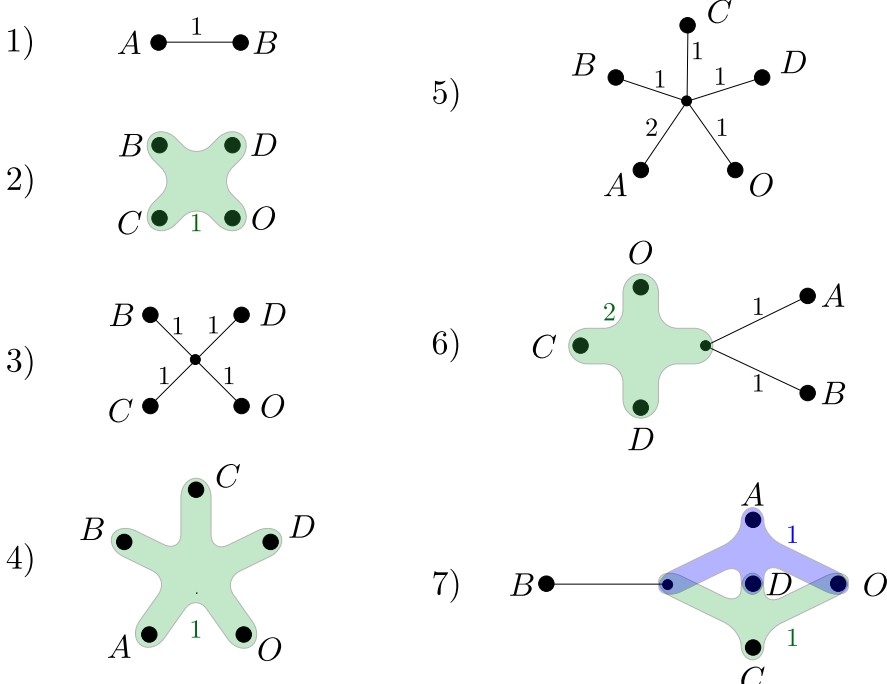

Figure 4: Hypergraphs corresponding to each symmetry orbit of extreme rays of the 4-party QLR cone, which notably coincides with the stabilizer cone. We have organized them by their labels as enumerated in [37] for the stabilizer cone. Similar to the holographic case, note that very few bulk vertices are needed – in fact, at most one. These are denoted by smaller, unlabeled points. As can be seen, some of the families of extreme rays admit realizations with no hyperedges, while others (such as the GHZ state) genuinely require hyperedges for their construction.

between the hypergraph and QLR cones for 5 parties[15].

### 3.3.1 The 4-Party Hypergraph Cone

With the preliminaries in place, we are now ready to pursue hypergraph instantiations of the extreme rays of the 4-party QLR cone defined by SA, SSA and Ingleton. Given that we have proven these inequalities on hypergraphs and that we have also found such hypergraph realizations for all extreme rays, we are able to provide a complete description of the hypergraph cone for 4 parties. Hypergraph representatives that realize these extreme rays are shown in Figure 4. One may easily verify that they precisely reproduce the entropies of the extreme rays of the 4-party stabilizer cone given in [37]. We hence conclude that the 4-party hypergraph cone and the 4-party stabilizer cone coincide.

### 3.3.2 The 5-Party Hypergraph Cone

We have been able to construct hypergraphs that realize representatives of every single one of the 162 orbits of extreme rays of the CLR cone for 5 parties, a complete description of which was found in [44]. While an explicit listing of these hypergraph realizations in the present paper would be rather cryptic and thus omitted, full details are available as supplemental

---

[15]Our proofs are partial because checking the contraction property on $i^k$ distance for all $k \leq R$ is computationally costly when $R$ is large. However, we are able to find contraction maps for all inequalities up to $k = 5$ and all but four up to $k = 6$. In all cases, we find agreement with conjecture 3.1.

material upon request. Here we simply note that the minimal number of bulk vertices needed to realize all of these extreme rays is always very small: 5 rays are realized by hypergraphs with zero bulk vertices, 27 with one, 105 with two and 25 with three, which is the largest number of bulk vertices required.

Since hypergraphs reach all extreme rays of the CLR cone and additionally violate classical monotonicity, we conclude that the hypergraph cone is strictly larger than the CLR cone. Note that all facets of the CLR cone except monotonicity may still be valid inequalities in consistency with our findings of extreme ray hypergraphs. This is indeed what seems to be true for all those facets. In Appendix A, we report on progress towards proving the 31 inequivalent 5-party QLR inequalities for hypergraphs[16]. The primary barrier to declaring all inequalities as proven happens to be the computational cost of checking the contraction property on $k'$-graphs with large $k' > k$ given a contraction map for some small $k$. Indeed, we observe that in practice it suffices to find a contraction map on $k$-graphs with $k = n = 5$. Then, one checks that the contraction property holds on higher ranks $k' > k$ too, as qualitatively expected from Corollary 3.1.2 and in suggestive agreement with Conjecture 3.1. Using Proposition 3.1.1, we have been able to conclusively prove 14 out of all 31 inequalities that define the QLR cone. Moreover, if one assumes the validity of Conjecture 3.1, our contraction maps also prove validity of 12 additional inequalities.

## 4 Hypergraphs as Quantum States

Up to this point, we have worked exclusively with entropy vectors, arguing that the discrete entropy of hypergraph models either does or does not obey certain inequalities. To the extent of what is known about the quantum entropy cone, this approach has allowed us to prove that the hypergraph entropy cone is not only contained in the former, but strictly inside. This is of course suggestive of the idea that the discrete entropy on hypergraphs really is computing the entropies of actual quantum states.

In this section, we will support this notion in a more direct fashion by providing a prescription for constructing quantum states from a given hypergraph, and conjecture that the resulting state faithfully realizes the exact same entropy ray as the original hypergraph. In doing so, we attempt to elevate hypergraphs from simply a diagrammatic representation of entropy vectors to a detailed encoding of physical quantum states with prescribed entropic properties. While we are as of yet unable to prove that the resulting state entropies always match the discrete entropies of the given hypergraph, we provide some physical motivation for our construction as well as perform non-trivial checks of it. Notably, the ingredients we utilize are all operations allowed for stabilizer states, further hinting that the hypergraph cone is intimately related to the stabilizer cone.

### 4.1 From Hypergraphs to Quantum States

The prescription for how to associate a quantum state to a given hypergraph is inspired by the tensor network formalism. The strategy consists of interpreting the hypergraph literally as a tensor network, where the rules that one associates to vertices and hyperedges are in principle unknown. In what follows, we provide a set of "Feynman rules" which, we conjecture, allows one to build a quantum state out of an arbitrary hypergraph whose subsystem von Neumann entropies match the hypergraph discrete entropies.

Given a hypergraph $G_K$ with $K = n + 1$ boundary vertices, a quantum state realization of

---

[16]The CLR cone consists of 33 inequalities, but one of them is monotonicity and two others are related by the purification symmetry of quantum mechanics, reducing the list of inequalities to 31 for the QLR cone.

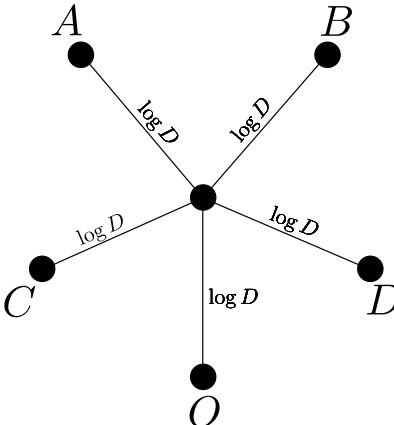

Figure 5: Graphical representation of an AME$(5, D)$ state. The local dimension of each of the parties is $D$. For $D = 2$, one could think of each of the parties as being a qubit. The key property of these AME states is that any subsystem looks maximally entangled. The given by 43 is reproduced when considering min-cuts of this graph since at most, half of the edges need to be cut. When combined with GHZ hypergraphs as in Figure 3, we can construct stabilizer states that reproduce a given entropy vector, indicating that all hypergraphs have a physical realization.

it will be written

$$|G_K\rangle = \mathcal{G}^{\mathcal{I}_1 \cdots \mathcal{I}_K} |\mathcal{I}_1, \ldots, \mathcal{I}_K\rangle, \tag{42}$$

where each index $\mathcal{I}_i$ ranges over some basis of the local Hilbert space to be associated to the $i^{\text{th}}$ party, and the Einstein summation convention is implicit. Because all the information about the state $|G_K\rangle$ is encoded in the tensor $\mathcal{G}^{\mathcal{I}_1 \cdots \mathcal{I}_K}$ of basis coefficients, the latter alone will be used to compactly specify a quantum state realization of a given hypergraph $G_K$.

### 4.1.1 Building Block Tensors

Observe that the graph structure of the neighborhood of any bulk vertex, meaning the vertex and the hyperedges attached to it alone, is just that of a star graph. In other words, each $k$-edge containing the given vertex may be thought of as a leg of the star between the vertex and the other $k-1$ vertices in the hyperedge. When considering discrete entropies given by min-cuts, these two pictures are equivalent under the inclusion or exclusion of the given vertex. Therefore one may think of an arbitrary hypergraph as a mosaic or network of star graphs glued together in some non-trivial way so as to reproduce the correct entropic behavior. This intuition motivates the search for a general quantum mechanical realization of general star graphs and the gluing mechanism thereof. In hindsight of the proposed strategy, we now introduce two ingredients which will be crucial in our construction of quantum states for a given hypergraph: absolutely maximally entangled (AME) states [46] to realize the local entanglement around bulk vertices and GHZ states [29] to capture the general behavior of a hyperedge.

A pure quantum state is said to be AME if it is maximally entangled for all bipartitions of the system. These states are organized into subclasses AME$(\Omega, D)$ of pure states on $\Omega$ parties, each having local Hilbert space dimension $D$. It is worth noting that while some such classes are empty, there is always some $D$ for which an $\Omega$-partite AME state exists[17] [48]. It immediately

---

[17]For example, there exists no quantum state in AME$(4, 2)$ [47], but a qutrit system can be built to realize a

follows from the definition that the von Neumann entropy $S(I)$ of a subsystem $I \subseteq [n]$ in an AME($\Omega, D$) state is given by

$$S(I) = \min\{|I|, \Omega - |I|\} \log D, \tag{43}$$

where the cardinality $|I|$ corresponds to the number of parties in $I$. The point of introducing AME states is that their entropies precisely match those of a star graph with $\Omega$ edges of uniform weight $\omega = \log D$. To see this, note that because a star graph just consists of a single bulk vertex and 2-edges connecting it to each of the $K$ boundary vertices, one need only consider two candidates to minimal cuts for a given $I \subseteq [\Omega]$: one that includes the bulk vertex, and one that does not. Their respective cut weights are $|I|\omega$ and $(\Omega - |I|)\omega$, thus demonstrating the agreement with (43) under the min-cut prescription in (4).

In general, the star graph that reproduces the local entropic structure of an arbitrary bulk vertex will be one of non-uniform weights, and therefore the association of one edge to each party would not immediately yield the entropies of an AME state. It is however a simple matter to bring such a star graph into a form suitable for the use of AME states as well. One just needs to think of a star graph that has $\ell$ edges of weights $\omega_i$ and total weight $\Omega = \sum_{i=1}^{\ell} \omega_i$ as arising from a uniform star with $\Omega$ edges of unit weight grouped together in parallel into $\ell$ sets of $\omega_i$ edges[18]. By associating a state in AME($\Omega, D$) to the latter and coloring its $\Omega$ subsystems non-injectively by $i = 1, \ldots \ell$ as dictated by the edge groupings, one precisely gets the desired entanglement structure among colored parties[19].

A rank-$\Omega$ tensor that corresponds to a state in AME($\Omega, D$) will be denoted by $T^{I_1 \cdots I_\Omega}$, with every tensor index $I_i = 1, \ldots, D$. If $\Omega$ is even, and if it exists for the given $D$, then $T^{I_1 \cdots I_\Omega}$ is also known as a *perfect tensor*. These tensors are defined as follows: consider a Hilbert space $\mathcal{H}$ and an arbitrary bipartite factorization $\mathcal{H} = \mathcal{H}_A \otimes \mathcal{H}_B$. Then a perfect tensor $T$ can be viewed as a map $\mathcal{H}_A \to \mathcal{H}_B$ which is an isometry $T^\dagger T = I_A$ for any relevant choice of bipartition with $|\mathcal{H}_A| \leq |\mathcal{H}_B|$. Such tensors have gained prominence in recent years due to their appearance in tensor networks and error correcting codes that serve as toy models of holography [49–51]. These tensor networks capture the leading order structure of the entanglement entropy of holographic systems, namely that they obey the RT formula, and so it is perhaps not surprising that they should appear in our attempt to convert hypergraphs to quantum states. As an explicit state vector, any AME($\Omega, D$) state can be written in the form

$$|\text{AME}(\Omega, D)\rangle = \frac{1}{\sqrt{d^s}} \sum_{k \in \mathbb{Z}_D^m} |k_1\rangle_{A_1} \ldots |k_m\rangle_{A_m} |\phi(k)\rangle_B, \tag{44}$$

for some bipartition of the system into $A = \{A_1, \ldots, A_m\}$ and $B = \{B_1, \ldots, B_{\Omega-m}\}$ with $m \leq \Omega - m$ and $\langle \phi(k) | \phi(k') \rangle = \delta_{kk'}$. Then the AME condition is the statement that the state in question can be written in the above form for *any* bipartition $A, B$ with $2m \leq \Omega$. We note here that this abstract representation of an AME state does not fix its exact form: for instance, the application of any set of $\Omega$ local unitaries acting on each of the subsystems would leave the form of (44) unchanged. We will remain agnostic as to which choice of local basis is the correct one and comment on the lack of uniqueness in our construction of states from hypergraphs shortly.

The introduction of GHZ states to account for hyperedges is motivated by the observation made in section 3 that a $\Omega$-partite qu-$D$-it GHZ state is entropically equivalent to a hypergraph with a single $\Omega$-edge of weight $\omega = \log D$ (see Figure 3). The tensor representation of any such state will be denoted by $\tilde{T}^{J_1 \cdots J_\Omega}$, with every tensor index $J_i = 1, \ldots, D$. It is straightforward to

---

4-partite AME, implying that AME(4, 3) is non-empty [48].

[18]Here, as will be done henceforth, weights are assumed rational such that graph weights are rescalable to integer numbers while preserving entropy rays.

[19]Different parties may end up with different local Hilbert space dimension, which will be given by $D^{\omega_i}$.

write this down for arbitrary $\Omega$ and $D$ as

$$\tilde{T}^{J_1,\cdots,J_\Omega} = \frac{1}{\sqrt{D}}\delta^{J_1\cdots J_\Omega} \qquad \text{where} \quad \delta^{J_1\cdots J_\omega} := \begin{cases} 1 & \text{if} \quad J_1 = \cdots = J_\Omega, \\ 0 & \text{otherwise.} \end{cases} \tag{45}$$

The explicit state vector in the computational basis is

$$|\text{GHZ}(\Omega,D)\rangle = \frac{1}{\sqrt{D}}\sum_{i=0}^{D-1}|i\rangle_1\ldots|i\rangle_\Omega\,. \tag{46}$$

As defined, one may consider the case $\Omega = 2$ as corresponding to a Bell pair between two qu-$D$-its, which takes the form of the $D$-dimensional identity matrix $\mathbb{1}_D$, up to a normalizing prefactor $1/\sqrt{D}$. Additionally, one notices that GHZ states of arbitrary local dimension $D$ are AME$(\Omega,D)$ states for both $\Omega \in \{2,3\}$, but are no longer so for any $\Omega \geq 4$.

### 4.1.2 Hypergraph Feynman Rules

Since the hypergraph entropy cone is polyhedral, we will focus on hypergraphs with rational weights, so that there always exists a finite scaling factor that allows us to uniformly scale all weights to be integer-valued[20]. Starting with the rescaled hypergraph, one performs a simple graph operation which can be easily seen to preserve all entropies while bringing the hypergraph to a more convenient form. This operation consists of replacing every hyperedge of weight $\omega$ by as many parallel hyperedges of unit weight, where by parallel we mean that each of the latter $\omega$ hyperedges of unit weight consists of precisely the same subset of vertices as the former. This manipulation sets the stage for a direct application of AME states to account for the local entanglement structure of bulk vertices.

To make sense of the tensor network interpretation of hypergraphs, one not only needs tensors, but also a prescription for contracting their indices. By associating AME tensors to bulk vertices and GHZ tensors to hyperedges, the first part of this challenge is mostly resolved. To attack the second part, note that a $k$-edge may be thought of as some virtual *node* consisting of *$k$ legs* between the node and each vertex[21]. To each leg one may assign a different index of the GHZ tensor. Then, in a general situation in which a hyperedge meets a bulk vertex, one would contract the appropriate GHZ tensor index with whichever AME tensor index corresponds to that edge from the perspective of the vertex. For concreteness, here we summarize the rules for building the tensor $\mathcal{G}^{\mathcal{I}_1\cdots\mathcal{I}_K}$ for a given hypergraph $G_K$:

1. At each bulk vertex of degree $d$ insert a rank-$d$ AME tensor $T^{I_1\cdots I_d}$, and assign a different tensor index to each one of the hyperedges containing it.

2. At each hyperedge of cardinality $k$ insert a rank-$k$ GHZ tensor $\tilde{T}^{J_1\cdots J_k}$, and assign a different tensor index to each one of the vertices it contains.

3. To every pair $s = (e,v) \in E \times V$ such that $v \in e$ and $v \notin \partial V$, there correspond two tensor indices: $I(s)$ assigned to $e$ for containing $v$ (rule 1) and $J(s)$ assigned to $v$ for being contained in $e$ (rule 2). These two indices are contracted across a Hadamard matrix $H$ of the appropriate order.

4. To every pair $s = (e,v) \in E \times V$ such that $v \in e$ and $v \in \partial V$, there correspond a single tensor index $J(s)$ assigned to $v$ for being contained in $e$ (rule 2). This index is added to the collective $i^{\text{th}}$ party index $\mathcal{I}_i$ of color $i = b(v)$.

---

[20]Note that this only causes a scalar rescaling of its entropy vector, thus leaving the ray itself invariant.

[21]Because these are artificial notions which do not belong to the graph, note the use of the words *node* instead of *vertex*, and *leg* instead of *edge*.

One may worry that the contraction of indices $I(s) = 1, \ldots, D_I$ and $J(s) = 1, \ldots, D_J$ instructed by step 3 may be ill-defined if their ranges happen to disagree, i.e. if $D_I \neq D_J$. Because $\Omega$-partite GHZ tensors (45) exist for any $D$, the subtlety lies on how to choose $D$ in a compatible way across all AME tensor insertions. In fact, one just has to choose a value for $D$ such that the AME$(d, D)$ class is non-empty for all degrees $d$ of bulk vertices in the hypergraph[22]. Then, all tensor insertions performed following rules 1 and 2 can be chosen to have local Hilbert space dimension $D$, guaranteeing compatibility in rule 3 across Hadamard matrices of order $D$.

In the examples we considered, the actual choice of local basis for the realization of the tensor $T$ in rule 1 was important for reproducing the correct entropy vector. As mentioned below (44), we were unable to find a universal rule for which specific representative tensor of a given AME class one should employ, but we observed that for each graph we considered, there was always some consistent choice that yielded a state with the desired entropies. We therefore believe that an appropriate completion of rule 1 that fixes this ambiguity should exist and formulate the following conjecture:

**Conjecture 4.1.** *There exists an appropriate refinement of rule 1 such that the entropies of all subsystems of the $K$-party quantum state $|G_K\rangle$ constructed from a hypergraph $G_K$ as described above agree precisely with the entropies obtained via the min-cut prescription applied to $G_K$, up to a common scaling factor.*

An analytical argument for this conjecture remains elusive, but explicit computations of all hypergraphs so far have indeed yielded states whose subsystem entropies precisely match the hypergraph discrete entropies. We provide some examples in the next section.

The connection to stabilizer states is also manifest in this procedure. Assuming that the AME state inserted at every bulk vertex is a stabilizer state [52–54], the state generated by these rules will be a contraction of stabilizer states (the stabilizer AME and GHZ states) and a stabilizer gate (Hadamard), and hence the resulting state will either be a stabilizer state or vanish. The validity of this conjecture would then immediately imply that hypergraph states constructed in this fashion are stabilizer states, and therefore the hypergraph entropy cone would be a subset of the stabilizer entropy cone. Note, however, that entropy vectors do not uniquely specify quantum states; provided Conjecture 4.1 is correct, the method above would produce a state with the correct subsystem entropies from any hypergraph, but it would not necessarily generate every single stabilizer state within the hypergraph cone.

As a final remark, we note the connection between the construction above and holographic tensor networks. Such tensor networks may be thought of as 2-graphs, in which case rule 2 on standard edges reduces to the insertion of maximally entangled rank-$d$ Bell pairs. Choosing their tensors to be $d$-dimensional identity matrices, the index contraction dictated by rule 3 of two symmetric Hadamard matrices accross them then trivializes to the standard identity map contraction of vertex tensor indices accross 2-edges. The upshot is a network of AME tensors of uniform local dimension joined together across edges via direct index contractions. In cases where the AME tensors are perfect tensors one then recovers the holographic tensor networks appearing in [49], which themselves are special cases of the more general random stabilizer tensor networks in [50, 51] obtained by replacing the perfect tensor or AME states with random stabilizer states.

---

[22]A simple way to see that this is always possible is to pick an AME$(d, D_d)$ state of arbitrary $D_d$ for each degree-$d$ bulk vertex, and then insert a tensor product of $L(D)/D_d$ copies of each at rule 1, where $L(D)$ is the lowest common multiple of all $D_d$ considered.

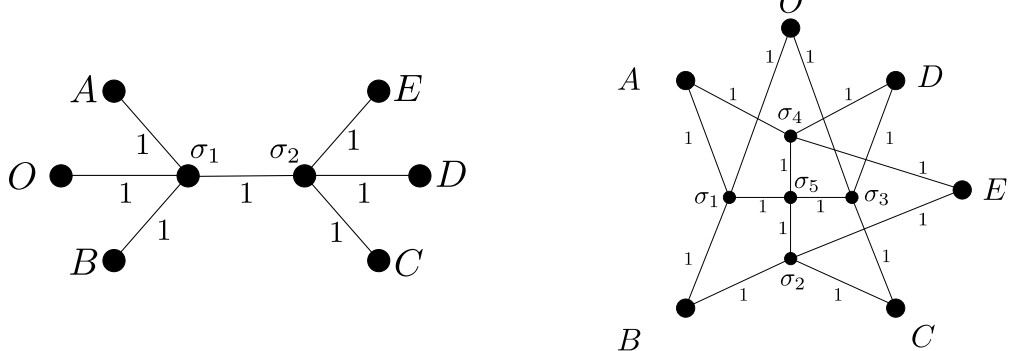

Figure 6: Two of the graphs that realize extreme rays of the 5-party holographic entropy cone. Boundary vertices are labeled by their party while bulk vertices are denoted by $\sigma_i$. Adapted from [21], these are rays 8 and 12 from left to right.

## 4.2 Explicit Constructions of Hypergraph States

As a warm-up, consider first the construction of standard 2-graph states, such as those which realize the extreme rays of the holographic entropy cone. The simplest of these which is not just a star graph (and thus realizable merely by a single AME state) is shown on the left of Figure 6 (graph 8 in Figure 1 of [21])[23]. As per rule 1, this graph requires the insertion of two 4-partite perfect tensors which, as remarked at the end of the previous section, should just be directly contracted along the 2-edge that connects the two bulk vertices.

The lowest-dimensional realization of a 4-partite perfect tensor is a 4-qutrit state whose simplest tensor representation reads[24]

$$T^{ijkl} = \frac{1}{3}\,\delta^{k,i\oplus j}\delta^{l,i\oplus 2j}, \tag{47}$$

where $x \oplus y := x + y \pmod 3$. Contracting two copies of this tensor on one respective index, the desired tensor representation of a quantum state for this ray is

$$\mathcal{G}_{R_8}^{\mathcal{OABCDE}} = \sqrt{3}\,T^{oabk}T^{cde}{}_k, \tag{48}$$

where in this case the collective indices are simply given by $\mathcal{X} = \{x\}$. For concreteness, the quantum state associated to $\mathcal{G}_{R_8}$ may be written out explicitly as

$$\left| G_{R_8} \right\rangle = \frac{1}{3\sqrt{3}} \sum_{i,j,m,n=0}^{2} \delta^{i\oplus 2j, m\oplus 2n} |i\rangle_O |j\rangle_A |i \oplus j\rangle_B |m\rangle_C |n\rangle_D |m \oplus n\rangle_E. \tag{49}$$

One then easily verifies that $\left| G_{R_8} \right\rangle$ gives precisely the desired entropies corresponding to ray 8 in Table 3 of [21].

As a less trivial holographic example, consider now the construction of a quantum state tensor $\mathcal{G}_{R_{12}}$ for the graph on the right of Figure 6 (graph 12 in Figure 1 of [21]) . Every bulk vertex in this graph has four edges of unit weight attached to it, corresponding again to the insertion of 4-partite perfect tensors on each. After contracting their indices as instructed by the edges structure, one is left with 12 free indices which are pairwise assigned to collective indices for each party as dictated by rule 4. Using (47) for bulk vertices, one obtains

$$\mathcal{G}_{R_{12}}^{\mathcal{ABCDEO}} = 9\,T^{i_1 i_2 i_3 i_4} T_{i_1}{}^{a_1 b_1 c_1} T_{i_2}{}^{d_1 e_1 o_1} T_{i_3}{}^{a_2 b_2 c_2} T_{i_4}{}^{d_2 e_2 o_2}, \tag{50}$$

---

[23]We thank Michael Walter for the idea to use tensor networks to reproduce the entropies of this graph.

[24]As shown in section 4.1.1, note that this state realizes the entropies of graph 2 in Figure 1 of [21].

where the index labels suggestively denote which collective boundary index each qutrit should be associated to via $\mathcal{X} = \{x_1, x_2\}$. One can check that this state precisely reproduces the desired entropy vector as well (cf. ray 12 in Table 3 of [21]).

For a simple yet interesting hypergraph state construction, we will look at the one in Figure 7, which realizes an extreme ray of the CLR cone for 5 parties. This hypergraph contains one bulk vertex and three hyperedges (one of them a 2-edge), all of unit weight. The required tensors for the 3-edge and the 5-edge are explicitly given by (45) with $D = 2$ for $\Omega = 3$ and $\Omega = 5$, respectively. The bulk vertex has only degree 3, for which a simple choice of AME representative is given by a 3-partite qubit GHZ state[25]. Applying the Hadamard contraction across tensor indices, the desired state tensor can be written

$$\mathcal{G}_{\text{CLR}_5}^{\mathcal{ABCDEO}} = 2\,\tilde{T}^{cedo_1 j_\sigma} H_{j_\sigma j_h}\,T^{j_h a i_h} H_{i_h i_\sigma}\,\tilde{T}^{i_\sigma b o_2}, \tag{51}$$

where $\mathcal{X} = \{x\}$ for all collective indices except for $\mathcal{O} = \{o_1, o_2\}$, and indices $i$ and $j$ have been employed for tensors associated to the 3-edge and 5-edge, respectively. Explicitly, as a state,

$$
\begin{aligned}
\left| G_{\text{CLR}_5} \right\rangle &= 2 \times \frac{1}{\sqrt{8}}\, \delta^{cedo_1 j_\sigma} H_{j_\sigma j_h}\, \delta^{j_h a i_h} H_{i_h i_\sigma}\, \delta^{i_\sigma b o_2}\, |a; b; c; d; e; o_1 o_2\rangle \\
&= \frac{1}{\sqrt{2}} \sum_{a,k,l=0}^{1} \delta^{k j_\sigma} H_{j_\sigma j_h}\, \delta^{j_h a i_h} H_{i_h i_\sigma}\, \delta^{i_\sigma l}\, |a; l; k; k; k; k\,l\rangle_{\text{ABCDEO}} \\
&= \frac{1}{\sqrt{2}} \sum_{a,k,l=0}^{1} H^{ka} H^{al}\, |a; l; k; k; k; k\,l\rangle_{\text{ABCDEO}}.
\end{aligned}
\tag{52}
$$

Written out,

$$
\begin{aligned}
\sqrt{8}\left| G_{\text{CLR}_5} \right\rangle = {}& |0000000\rangle + |0011110\rangle + |0100001\rangle + |0111111\rangle \\
& + |1000000\rangle - |1011110\rangle - |1100001\rangle + |1111111\rangle,
\end{aligned}
\tag{53}
$$

where, in order, the qubits correspond to $A$, $B$, $C$, $D$, $E$, $O_1$ and $O_2$. Once again, the entropies of this state match exactly those of the entropy ray obtained by the min-cut prescription applied to the hypergraph in Figure 7, whose computation we leave as an exercise for the reader.

## 5 Discussion and Future Directions

We have shown that hypergraphs are able to capture a larger entropy cone than graphs, thus strictly containing the holographic entropy cone, while remaining consistent with universal quantum inequalities. In particular, for up to 4 parties, we have found that the hypergraph entropy cone is equivalent to the stabilizer and quantum linear rank cones. In the 5-party case, we obtained compelling evidence that hypergraph entropies continue to obey and tightly saturate all quantum linear rank inequalities. Assuming hypergraphs indeed correspond to physical quantum states, our results suggest that hypergraphs can be a powerful tool to discover and prove novel entropy inequalities, as well as construct explicit quantum states that realize a given entropy vector.

In the spirit of extending the graph tools developed in [20] to study more general classes of entanglement structures, one may wonder if hypergraphs could themselves admit an additional upgrade. Going beyond hypergraphs could in principle provide a combinatorial tool

---

[25]Note that this is no longer the case for a $k$-partite GHZ with $k \geq 4$. In particular, the 3-partite case is realizable holographically by a star graph, which is the reason why the star associated to this bulk vertex admits a GHZ tensor type of realization, otherwise used only for hyperedges. Higher-party GHZ states are not holographic, and thus also not realizable by star graphs at all.



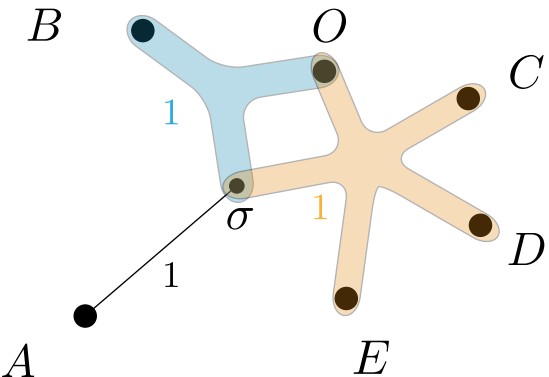

Figure 7: A hypergraph that realizes one of the extreme rays of the 5-party CLR cone. Boundary vertices are labeled by their party while the only bulk vertex is denoted by $\sigma$.

to explore the full quantum entropy cone, regions of which are inaccessible to hypergraphs already at 4 parties. A construction that breaks the Ingleton bound might also shed light on whether there exist genuinely quantum 5-partite states (i.e. with non-monotonic entropies) that violate the Ingleton inequality [28, 37].

Below, we outline several other avenues of future study related to hypergraphs.

## 5.1 The Hypergraph Cone and Quantum States

Perhaps the most pressing issue is the question of realizability of all hypergraph entropies using quantum states. We have made partial progress in this direction through proving the validity of many of the QLR inequalities (including SA and SSA), as well as proposing a method to explicitly write down a quantum state realizing the hypergraph entropies.

In section 3, we showed that the hypergraph and QLR cones are identical for up to 4 parties by demonstrating that all 4-party QLR inequalities are valid on hypergraphs (which implies containment), and also that all extreme rays of the QLR cone are hypergraph-realizable (which implies tightness). Additionally, in appendix A we exhibit contraction maps which prove several of the QLR inequalities for 5 parties. In fact, we believe they are sufficient to prove all of them, but are limited by the computational cost of checking the contraction property for large hypergraph ranks. In particular, we have not found a failed contraction map as of yet for any linear rank inequality for 5 parties or any evidence of non-convergence of the proof technique. We are thus compelled to make the following conjecture:

**Conjecture 5.1.** *The n-party hypergraph and QLR cones coincide for all n.*

It is known that the CLR cone is strictly inside the classical cone and thus the quantum cone [43]. It is unclear if the replacement of monotonicity by weak monotonicity in the CLR cone, which defines the QLR cone, would allow for the latter to reach beyond the quantum cone. We believe this is very unlikely[26] and thus validity of Conjecture 5.1 would imply that the hypergraph cone indeed encodes physically meaningful quantum state entropies.

On a more constructive direction, the prescription proposed in section 4 also suggests that hypergraph entropies precisely correspond to those of a specific subset of quantum states. This is already manifestly true at 4-parties, in which case we have shown that the hypergraph

---

[26]For instance, quantum states are able to violate all QLR inequalities that are not SA and SSA.

and stabilizer entropy cones are equivalent since the latter coincides with the 4-party QLR cone. For larger party number, it is known that stabilizer states obey all balanced linear rank inequalities based on common information [37], but it is unclear if they fill up this cone or even if they satisfy other more general linear rank inequalities that arise for $n \geq 6$ [55]. Therefore, although little is known about the stabilizer entropy cone in terms of extreme rays already for 5 parties, it is known that these rays all lie inside the QLR cone. Appendix A provides strong evidence to support that this property is shared by hypergraphs as well, and motivates the following conjecture:

**Conjecture 5.2.** *The n-party hypergraph and stabilizer cones coincide for all n.*

An open question in quantum Shannon theory is whether the stabilizer and QLR cones are the same for more than 4 parties. In the case that the *n*-party stabilizer cone is equivalent to the *n*-party QLR cone, the two conjectures above naturally collapse to one.

If the hypergraph cone really is inside the quantum cone, the machinery developed in this paper would provide an efficient way of computing entanglement entropies[27] of some class of quantum states (such as stabilizer states if Conjecture 5.2 holds) and a proof method for the inequalities they obey. The ability to encode the entanglement structure of a given quantum state in a hypergraph could then serve as a partial witness for whether a state belongs to that class[28].

Naturally, one could consider proving equality of all three of the hypergraph, stabilizer, and QLR cones. Section 4 provides a construction possibly leading to equivalence of the first two, which would prove Conjecture 5.2. As for Conjecture 5.1, it might be conceivable to prove that hypergraph states obey all QLR inequalities with our methods. However, proving that hypergraphs completely fill the QLR cone would require a better understanding of the connection between hypergraphs and linear ranks.

**Note added.** Shortly after the completion of this work, it was shown in [56] that all hypergraph entropy vectors can be achieved by stabilizer tensor networks, hence proving containment of the hypergraph cone in the stabilizer cone. Soon after, it has been recently shown that there exists a 5-party (6-qubit) stabilizer state which violates an entropy inequality that must hold for all 5-party hypergraphs [57]. This proves that the hypergraph is strictly contained within the stabilizer cone, and hence that both conjectures above are false.

## 5.2 Searching for New Hypergraph Inequalities

Thus far we have not put a premium on discovering new inequalities for hypergraph entropies, rather relying on proving old inequalities true for potentially equivalent mathematical constructs. However, it may be the case that the classes of objects we considered are not equivalent, as could happen starting at 5 parties for which the complete cone of stabilizer entropies is not known. Regardless, in these cases, one should be able to echo the procedure used in the holographic entropy cone to search for new inequality candidates that "slice off" extreme rays that are impossible to construct, while leaving already constructed ones intact, and to run the generalized contraction map algorithm on these new candidate inequalities. This is a promising future direction, though it may quickly become computationally intractable in the absence of better linear/semidefinite programming techniques. As such, we leave this to future work.

One possible direction forward is to utilize the attention paid to hypergraphs in machine learning [58] to aid in finding hypergraphs that realize extreme rays. It is quite possible that

---

[27]The complexity of computing entropies would reduce to that of computing min-cuts.

[28]Here we specify that it is a partial witness because there is a possibility that states that lie within the specified cone are not of the specified type (i.e. it is only a necessary condition).

machine learning techniques would allow deeper probes into higher party hypergraph entropy cones than the standard analytic approaches.

## 5.3 Non-Holographic Bit Threads

A version of the max-flow min-cut theorem known as the Menger property also holds for hypergraphs [59], where the minimal cut on hyperedge weights separating a source from a sink is equal to the maximal flow allowable from the source to the sink. The extension of the max-flow min-cut theorem to hypergraphs suggests a connection to the bit thread program of [15,60]. The key insight in that area was the connection of threads to RT surfaces via the max-flow min-cut theorem, in a manner that survives discretization of the bulk spacetime into graph form. Therefore, some analog of bit threads should survive for hypergraphs, and it would be interesting to study such properties in the context of quantum states or tensor networks. While entropy inequalities appeared harder to prove in the thread formalism than with cuts [16,60], they also usually offered some deeper insight into the structure and properties of holographic states. It is conceivable that some analogous lessons about hypergraph states may be drawn from a thread reformulation of our analysis.

One may then ask about the relationship between hypergraph techniques and the entanglement of purification conjectures of [30, 31] (and their multipartite and conditional extensions [32–34]), taking advantage of the their connection to bit threads [61–63]. In this framework, the generalized contraction map can potentially be extended to study inequalities related to the entanglement of purification for hypergraph states, assuming an appropriate generalization of an entanglement wedge cross section as a partial cut of a hypergraph. This generalization, for example, could describe a hypergraph for which some of the vertices have been deleted, with hyperedges previously ending on those vertices rendered to be "dangling". The entanglement of purification question would then become one of how to complete the dangling edges to minimize some source-sink cut, in strong analogy to holographic entanglement of purification techniques.

## 5.4 Connections to Higher Derivative Gravity

It is possible that there is a connection between hypergraph states and ground or thermal states of boundary theories dual to higher derivative gravity theories. At least in certain situations [64], the entanglement entropy of a boundary subregion in these cases reduces to the Wald entropy, which in particular contains a topological component. It is tempting to represent this topological component as a hyperedge connecting all of the boundary subregions, and it would be worth investigating a potential connection with the techniques used in this paper.

## Acknowledgments

We thank David Avis, Sebastian Fischetti, Veronika Hubeny, Alex Maloney, Sepehr Nezami, Xiaoliang Qi, Djorde Radecevic, Mukund Rangamani, Massimiliano Rota and Michael Walter for useful comments and interesting discussions. N.B. is supported by the National Science Foundation under grant number 82248-13067-44-PHPXH, by the Department of Energy under grant number DE-SC0019380, and by New York State Urban Development Corporation Empire State Development contract no. AA289. S.H.C. is supported by "la Caixa" Foundation (ID 100010434) under fellowship LCF/BQ/AA17/11610002 and by the National Science Foundation under grant number PHY-1801805. V.P.S. gratefully acknowledges support by the NSF GRFP under Grant No. DGE 1752814.

# A  Proofs of Linear Rank Inequalities

Only the 24 genuinely 5-partite inequalities of the 5-party QLR are listed in Table 4, the other 7 corresponding to uplifted instances of SA, SSA and Ingleton, already proven for all hypergraphs in section 3.2.3. The search for a contraction map is carried out with the LHS written in its standard form, thus involving $L$ terms, while the RHS is written in expanded form ($\beta_i$ copies of $S_i$ with unit coefficient), thus involving $\beta_T = \sum_{i=1}^{R} \beta_i$ terms. Writing the LHS in standard form reduces the complexity of finding a contraction map by minimizing its domain, without affecting the efficacy of the proof method. On the other hand, expanding out the RHS maximizes the likelihood of finding a contraction map for a given valid inequality. Ideally, considering Proposition 3.1.1, one would like a RHS with the least possible number of terms, but this happens to be insufficient for some proofs. Basically, failure to find a contraction map with the RHS in standard form does not preclude the existence of a contraction map with the RHS in the expanded form. This already happens to be important for the proof of 5-partite holographic entropy inequalities, for which the standard form is insufficient in some cases. To be as general as possible, the choice here has been the conservative one of fully expanding out the RHS, at the cost of not been able to fulfill the conditions of Proposition 3.1.1 for some inequalities.

In most cases, these maps have been constructed by demanding the contraction property on hypergraphs of low rank $k \leq 4$. Quite generally, the $k$-distance for greater $k$ then turns out to automatically contract on those maps too. This agrees with the intuition drawn from corollary 3.1.2. More importantly, in all cases we find supporting evidence for Conjecture 3.1 that validity of a $k$-party inequality on $(k+1)$-graphs implies its validity on all hypergraphs. As a result, we expect the $k$-distance for all $k$ larger than those listed in the last column of Table 4 to already contract on their respective maps in Tables 5 to 28. The only obstruction we have found to performing such a check is its computational cost.

In the following tables, we provide a succinct representation of the proofs (i.e. contraction maps) for the 24 genuinely 5-partite inequalities of the 5-party QLR cone. We have encoded all relevant information compactly as follows. Note that an entropy inequality of the form 5 may alternatively be specified by the inward-pointing vector normal to the hyperplane in entropy space corresponding to the saturation of that inequality. In other words, the linear inequality 5 can be written as $Q \cdot S \geq 0$, where $S$ is an arbitrary entropy vector and $Q$ the inequality vector referred to above. The order of the entries of $S$ and $Q$ is determined by the standard lexicographic ordering of the power set of $n$ parties (excluding the empty set), namely $\{A, B, C, D, E, AB, AC, \ldots, ACDE, BCDE, ABCDE\}$. Each table below is captioned by a different inequality, specified in terms of the vector $Q$, and shows the corresponding contraction map compactly encoded as described next. Recall that the contraction map is a function from all possible $2^L$ bit strings of length $L$ to some set of bit strings of length $R$[29]. It can thus be uniquely defined by the image of every bit strings in the domain. The image bit strings, understood as digits of a binary number, can then be compactly expressed in their decimal-base representation. Ordering all inequality entropies lexicographically and the LHS domain bit strings in increasing binary order, an array of $2^L$ base-10 numbers can thus be used to uniquely specify the output of the contraction map on every input bit string (see the right-most column of Table 3 for an example of this notation). The following tables provide the integer arrays encoding the contraction map for each inequality, which should be read left-to-right first.

---

[29]In all cases below $R = \beta_T$, corresponding to the expanded form of the RHS.

Table 4: Summary of the status of the proof by contraction of QLR inequalities on hypergraphs. The numbering of inequalities corresponds to the labels used in Tables 5 to 28, which show those inequalities and their corresponding contraction map. Columns $L$ and $R$ are the number of terms on the LHS and RHS of the inequality in its standard form, respectively. Columns $\alpha_T$ and $\beta_T$ give the value of $\sum_{i=1}^{L} \alpha_i$ and $\sum_{i=1}^{R} \beta_i$, respectively. The last column lists the largest hypergraph rank $k$ for which the contraction property of the $k$-distance has been checked on the maps given below. Inequalities for which this check has reached $k = \beta_T$ are valid for all hypergraphs by Proposition 3.1.1 (underlined and bold), and those for which $k \geq 6$ are plausibly valid by Conjecture 3.1 (bold).

| Inequality | $L$ | $R$ | $\alpha_T$ | $\beta_T$ | Checked $k$ |
|---:|---|---|---|---|---|
| **1** | 6 | 6 | 6 | 6 | 6 |
| **2** | 6 | 6 | 6 | 6 | 6 |
| **3** | 6 | 6 | 6 | 6 | 6 |
| 4 | 6 | 7 | 8 | 8 | 7 |
| 5 | 7 | 7 | 8 | 8 | 7 |
| **6** | 7 | 7 | 7 | 7 | 7 |
| **7** | 7 | 7 | 7 | 7 | 7 |
| 8 | 7 | 7 | 8 | 8 | 7 |
| **9** | 7 | 7 | 7 | 7 | 7 |
| 10 | 6 | 7 | 8 | 8 | 7 |
| **11** | 7 | 7 | 7 | 7 | 7 |
| 12 | 7 | 8 | 10 | 10 | 6 |
| 13 | 8 | 8 | 9 | 9 | 6 |
| 14 | 8 | 8 | 8 | 8 | 6 |
| 15 | 9 | 8 | 10 | 10 | 6 |
| 16 | 8 | 8 | 8 | 8 | 6 |
| 17 | 7 | 8 | 10 | 10 | 6 |
| 18 | 8 | 8 | 8 | 8 | 6 |
| 19 | 8 | 8 | 9 | 9 | 6 |
| 20 | 9 | 8 | 10 | 10 | 6 |
| 21 | 9 | 9 | 13 | 13 | 4 |
| 22 | 7 | 9 | 10 | 10 | 4 |
| 23 | 9 | 9 | 13 | 13 | 4 |
| 24 | 7 | 9 | 10 | 10 | 4 |

Table 5: $Q_1 = \{0,-1,0,-1,0,1,-1,1,0,1,1,0,0,0,0,0,-1,0,0,1,0,0,-1,0,1,0,0,0,-1,0,0\}$

| 0 | 1 | 1 | 3 | 4 | 5 | 0 | 1 | 2 | 3 | 3 | 11 | 6 | 7 | 2 | 3 | 1 | 5 | 5 | 1 | 5 | 21 | 1 | 5 | 0 | 1 | 1 | 3 | 4 | 5 | 0 | 1 |
|---|---|---|---|---|---|---|---|---|---|---|---|---|---|---|---|---|---|---|---|---|---|---|---|---|---|---|---|---|---|---|---|
| 4 | 0 | 5 | 1 | 6 | 4 | 4 | 0 | 6 | 2 | 7 | 3 | 38 | 6 | 6 | 2 | 5 | 1 | 13 | 5 | 4 | 5 | 5 | 1 | 4 | 0 | 5 | 1 | 6 | 4 | 4 | 0 |

Table 6: $Q_2 = \{0,-1,0,0,-1,1,-1,0,0,0,0,1,0,1,0,0,0,0,1,0,1,1,0,-1,0,-1,0,0,-1,0,0\}$

| 0 | 2 | 1 | 3 | 1 | 3 | 3 | 7 | 1 | 3 | 5 | 7 | 3 | 11 | 1 | 3 | 4 | 6 | 5 | 7 | 0 | 2 | 1 | 3 | 5 | 7 | 21 | 5 | 1 | 3 | 5 | 7 |
|---|---|---|---|---|---|---|---|---|---|---|---|---|---|---|---|---|---|---|---|---|---|---|---|---|---|---|---|---|---|---|---|
| 2 | 6 | 3 | 7 | 3 | 7 | 11 | 3 | 0 | 2 | 1 | 3 | 1 | 3 | 3 | 3 | 6 | 38 | 7 | 6 | 2 | 6 | 3 | 7 | 4 | 6 | 5 | 7 | 0 | 2 | 1 | 3 |

Table 7: $Q_3 = \{0,-1,0,0,0,1,-1,0,0,1,0,0,0,0,-1,0,0,0,1,0,1,-1,0,1,1,0,0,-1,-1,0,0\}$

| 0 | 1 | 2 | 3 | 1 | 3 | 3 | 11 | 1 | 5 | 3 | 7 | 3 | 7 | 7 | 15 | 4 | 5 | 6 | 7 | 0 | 1 | 2 | 3 | 5 | 21 | 7 | 5 | 1 | 5 | 3 | 7 |
|---|---|---|---|---|---|---|---|---|---|---|---|---|---|---|---|---|---|---|---|---|---|---|---|---|---|---|---|---|---|---|---|
| 2 | 0 | 6 | 2 | 3 | 1 | 7 | 3 | 3 | 1 | 7 | 3 | 19 | 3 | 3 | 7 | 6 | 4 | 38 | 6 | 2 | 0 | 6 | 2 | 7 | 5 | 6 | 7 | 3 | 1 | 7 | 3 |

Table 8: $Q_4 = \{-2,-2,0,0,0,2,1,1,1,1,1,1,0,0,0,-1,-1,-1,0,0,0,0,0,0,-1,0,0,0,0,0\}$

```
0   2   4   6   8  10  12  14   2   3   0   2   0   2   4   6   4   0   5   4   0   2   4   6   6   2   4   0   2   0   0   2
8   0   0   2   9   8   8  10  10   2   2   0   8   0   0   2  12   4   4   0   8   0   0   2  14   6   6   2  10   2   2   0
6  14  14  30  14  30  30  62  14   6   6  14   6  14  14  30  14   6   6  14   6  14  14  30  78  14  14   6  14   6   6  14
14  6   6  14  10  14  14  30  78  14  14   6  14   6   6  14  78  14  14   6  14   6   6  14 206  78  78  14  78  14  14   6
```

Table 9: $Q_5 = \{-1,-2,0,0,0,2,-1,1,1,1,1,1,0,0,-1,0,-1,-1,0,0,0,0,0,0,1,0,0,0,0,-1,0\}$

```
0   1   1   3   1   5   3   1   1  17   3   1   5   1   7   3   2   3   3  11   0   1   1   3   0   1   1   3   1   1   3   1
4   5   0   1   5  13   1   5   0   1   1   1   1   5   3   1   6   7   2   3   4   5   0   1   2   3   0   1   0   1   1   0
6   2   7   3   7   3  39   7   7   3  39   7  39   7 103  39  22   6   6   2   6   2   7   3   6   2   7   3   7   3  39   7
22  6   6   2   6   7   7   3   6   2   7   3   7   3  39   7 150  22  22   6  22   6   6   2  22   6   6   2   6   2   7   3
```

Table 10: $Q_6 = \{-1,0,-1,-1,0,1,1,1,1,1,0,-1,1,0,1,-1,0,0,-1,0,-1,0,0,0,0,0,0,0,0,0\}$

```
0   1   2   3   4   0   6   2   1   9   0   1   0   1   2   0   1   3   3  19   0   1   2   3   3   1   1   3   1   0   0   1
2   0   6   2   6   2  38   6   3   1   2   0   2   0   6   2   3   1   2   3   2   0   6   2   7   3   3   1   3   1   2   0
4   0   0   1  12   4   4   0   5   1   1   0   4   0   0   0   5   1   1   3   4   0   0   1   7   3   3   1   5   1   1   0
6   2   2   0   4   0   6   2   7   3   3   1   6   2   2   0   7   3   3   1   6   2   2   0  71   7   7   3   7   3   3   1
```

Table 11: $Q_7 = \{0,-1,-1,-1,0,1,1,1,-1,1,1,0,0,1,1,-1,-1,0,0,0,0,0,0,0,-1,0,0,0,0,0\}$

```
0   1   1   9   2   3   0   1   4   0   5   1   6   2   4   0   2   3   0   1   3  19   1   3   0   1   1   0   2   3   0   1
4   0   5   1   0   1   1   0   5   1  37   5   4   0   5   1   6   2   4   0   2   3   0   1   4   0   5   1   0   1   1   0
2   0   0   1   6   2   2   0   6   2   4   0  70   6   6   2   6   2   2   0   2   3   0   1   2   0   0   0   6   2   2   0
6   2   4   0   2   0   0   0   4   0   5   1   6   2   4   0  14   6   6   2   6   2   2   0   6   2   4   0   2   0   0   0
```

Table 12: $Q_8 = \{0,0,0,-1,0,0,0,1,0,-1,0,-1,0,0,0,1,-1,1,0,-1,0,1,2,1,1,0,-1,0,0,-2,0\}$

```
0   1   1   3   3   7   7  15   1   3   3  19   7  15  23   7   4   5   5   7   7  15  39  47   5   7   7  23  23   7  55  39
4   5   5   7   7  15  23   7   5   7   7  23  71  79  87  71  20  21  21  23  23   7  55  39  21  23  23   7  87  71 119 103
16 17  17  19   1   3   3   7  17  19  19 147   3   7   7   3  20  21  21  23   5   7   7  15  21  23  23  19   7   7  23   7
20 21  21  23   5   7   7   7  21  23  23  19   7  15  23   7  28  29  29  31  21   5  23   7  29  31  21  23  23   7  55  39
```

Table 13: $Q_9 = \{0,0,0,0,0,0,0,-1,0,-1,0,0,0,-1,0,1,1,0,0,1,1,1,1,-1,1,-1,-1,0,-1,0,0\}$

```
0  1   2   3   4   5   6   7   1   9   3  11   5  13   7  15   1   3   3  19   5   7   7  23   3  11  11  27   7  15  15  31
4  5   6   7  12  13  14  15   5  13   7  15  13  77  15  13   5   7   7   3  13   5  15   7   7  15  15  11   5  13   7  15
2  3   6   7   6   7  38  39   3  11   7   3   7  15   6   7   3   7   7  23   7  23  39  55   7   3   3  19   7   7   7  23
6  7  14  15  14  15  46  47   7  15  15   7  15  79  14  15   7   7  15   7  15   7  47  39  71   7   7   3   7  15  15   7
```

Table 14: $Q_{10} = \{0,0,0,0,0,0,0,0,-1,-1,-1,0,-1,0,0,1,1,0,1,0,0,2,1,1,1,-2,0,0,0,-2,0\}$

```
0   1   1   3   1   3   3 131   3   7   7  15   7  15  15   7   4   5   5   7   5   7   7 135   7  15  15  31  15  31   7  15
4   5   5   7   5   7   7 135   7  15  15  47  15   7  47  15  12  13  13  15  13   5  15   7  15  31  47  63   7  15  15  31
4   5   5   7   5   7   7 135   7  15  15  15  15  79  79  15  12  13  13  15  13  15  15 143  15  31  13  15  79  95  15  31
12 13  13  15  13  15  15 143  15  15  47  15  79  15 111  79 140 141 141 143 141  13 143  15  13  15  15  31  15  31  47  15
```

Table 15: $Q_{11} = \{0,0,0,0,0,0,0,0,0,-1,-1,0,0,0,-1,1,1,0,0,-1,1,1,1,1,1,1,-1,0,-1,0,-1,0\}$

```
0   1   1   3   1   9   3  11   1   5   3   7   5  13   7  15   2   3   3  19   3  11  11  27   3   7   7  23   7  15  15  31
2   3   3   7   3  11   7   3   3   7   7  39   7  15  39   7   6   7   7  23   7   3   3  19   7  23  39  55   7   7   7  23
4   5   5   7   5  13   7  15   5  13   7  15  69  77  71  79   6   7   7   3   7  15  15  11   7  15   7   7   5  13   7  15
6   7   7   7   7  15  39   7   7  15  39   7  71  79 103  71  14  15  15   7  15   7   7   3  15   7   7  39   7  15  39   7
```

Table 16: $Q_{12} = \{-2,0,0,0,-2,2,1,1,1,-1,-1,2,0,1,1,0,0,-2,-1,0,0,1,0,0,0,0,0,0,-1,0\}$

```
0    1   1   3   1   3   5   1   5  13  13   5  13   5  77  13   4   0   5   1   5   1  13   5  13   5  77  13  77  13 205  77
2    3   3  19   0   1   1   3   1   5   5   1   5   1  13   5   6   2   7   3   4   0   5   1   5   1  13   5  13   5  77  13
2    3   0   1   3  35   1   3   1   5   5   1   5   1  13   5   6   2   4   0   7   3   5   1   5   1  13   5  13   5  77  13
6    2   2   3   2   3   0   1   0   1   1   0   1   0   5   1  14   6   6   2   6   2   4   0   4   0   5   1   5   1  13   5
12  13   4   5   4   5   0   1  29  61  13  29  13  29   5  13  14  12   6   4   6   4   4   0  13  29   5  13   5  13  13   5
14  15   6   7   6   7   2   3  13  29   5  13   5  13   1   5 270  14  14   6  14   6   6   2  12  13   4   5   4   5   5   1
14  15   6   7   6   7   2   3  13  29   5  13   5  13   1   5 270  14  14   6  14   6   6   2  12  13   4   5   4   5   5   1
270 14  14   6  14   6   6   2  12  13   4   5   4   5   0   1 782 270 270  14 270  14  14   6  14  12   6   4   6   4   4   0
```

Table 17: $Q_{13} = \{-1,-2,0,0,0,2,1,1,-1,1,1,1,-1,0,0,-1,-1,0,0,0,1,0,0,-1,1,0,0,0,-1,0,0\}$

```
 0  1   1   3  2  3  3 35  2  3  3  7  6  7  7  3  8  9  0  1 10 11  2  3 10 11  2  3  14 15  6  7
 1  3   5   7  0  1  1  3  3  7  7 23  2  3  3  7  0  1  1  5  2  3  0  1  2  3  3  7   6  7  2  3
 1  9   9   1  0  1  1  3  0  1  1  3  2  3  3  1  9 25  1  9  8  9  0  1  8  9  0  1  10 11  2  3
 9  1  13   5  1  1  5  1  1  3  5  7  0  1  1  3  1  9  5  1  0  1  1  0  1  1  5  2   3  0  1
12  4  13   5 14  6 15  7 14  6 15  7 78 14 14  6 14 12 12  4 78 14 14  6 78 14 14  6 206 78 78 14
13  5  45  37 12  4 13  5 15  7 13  5 14  6 15  7 12  4 13  5 14  6 12  4 14  6 15  7  78 14 14  6
13  5  45  13 12  4 13  5 12  4 13  5 14  6 15  7 12 13 13  5 14 12 12  4 14 12 12  4  78 14 14  6
45 13 301  45 13  5 45 13 13  5 45 13 12  4 13  5 13  5 45 13 12  4 13  5 12  4 13  5  14  6 12  4
```

Table 18: $Q_{14} = \{-1,0,-1,0,0,1,1,1,0,0,-1,0,1,0,-1,0,0,0,-1,-1,1,0,1,1,1,0,0,-1,0,-1,0\}$

```
 0 1 1  3 1  5 3  7   2  3  3 19  3  7  7 23 1  9  3 11  5 13  7 15  3 11 11 27  7 15 15 31
 2 3 3 11 0  1 1  3  10 11 11 27  2  3  3 19 3 11 11 27  1  9  3 11 11 27 27 59  3 11 11 27
 4 5 0  1 5 13 1  5   6  7  2  3  7  5  3  7 5 13  1  9 13 77  5 13  7 15  3 11  5 13  7 15
 6 7 2  3 4  5 0  1  14 15 10 11  6  7  2  3 7 15  3 11  5 13  1  9 15 11 11 27  7 15  3 11
 2 0 3  1 3  1 35 3   6  2  7  3  7  3  3  7 0  1  1  3  1  5  3  7  2  3  3 11  3  7  7 15
 6 2 7  3 2  0 3  1  14 10 15 11  6  2  7  3 2  3  3 11  0  1  1  3 10 11 11 27  2  3  3 11
 6 4 2  0 7  5 3  1  14  6  6  2 15  7  7  3 4  5  0  1  5 13  1  5  6  7  2  3  7  5  3  7
14 6 6  2 6  4 2  0 142 14 14 10 14  6  6  2 6  7  2  3  4  5  0  1 14 15 10 11  6  7  2  3
```

Table 19: $Q_{15} = \{-1,0,0,-2,-1,1,1,2,-1,-1,1,1,1,1,2,0,-1,0,-1,0,0,0,0,-1,-1,0,0,0,0,0,0\}$

```
  0  3  1 35  1   7  5  3  2 35  3 99  0  3  1 35  2 11  0  3  3  15  1   7 10  3  2 35  2  11 0  3
 12 15  4  7 13 143  5 15  4  7  0  3  5 15  1  7 14 143  6 15 15 399  7 143  6 15  2  7  7 143 3 15
  4  1  5  3  5   3 21  1  0  3  1 35  1  1  5  3  0  3  1  1  1   7  5   3  2  1  0  3  0   3 1  1
 44 13 12  5 12  15  4  7 12  5  4  1  4  7  0  3 12 15  4  7 13 143  5  15  4  7  0  3  5  15 1  7
  8  1  0  3  0   3  1  1 10  3  2 35  2  1  0  3 10  3  2  1  2   7  0   3 26 11 10  3 10   3 2  1
 44 13 12  5 12  15  4  7 12  5  4  1  4  7  0  3 12 15  4  7 14 143  6  15 14  7  6  3  6  15 2  7
 12  0  4  1  4   1  5  0  8  1  0  3  0  0  1  1  8  1  0  0  0   3  1   1 10  3  2  1  2   1 0  0
556 12 44  4 44  13 12  5 44  4 12  0 12  5  4  1 44 13 12  5 12  15  4   7 12  5  4  1  4   7 0  3
```

Table 20: $Q_{16} = \{0,-1,0,-1,0,1,-1,1,0,1,0,0,1,-1,0,0,0,0,0,1,-1,-1,1,1,1,0,-1,0,0,-1,0\}$

```
 0 1  1  9 1 3   3 11  2  3  3 11  3 19 11 27 1  5  5 13  3  7  7 15  3  7  7 15  7 23 15 31
 1 3  3  1 3 7   7  3  3  7  3  3  7 23  3 19 3  7  1  5  7 23  3  7  7 23  3  7 23 55  7 23
 8 9  9 13 0 1   1  9 10 11 11  9  2  3  3 11 9 13 13 77  1  5  5 13 11 15 15 13  3  7  7 15
 0 1  1  5 1 3   3  1  2  3  3  1  3  7  3  3 1  5  5 13  3  7  1  5  3  7  7  5  7 23  3  7
 2 0  3  1 3 1   7  3 10  2 11  3 11  3  3 11 0  1  1  5  1  3  3  7  2  3  3  7  3  7  7 15
 3 1  7  3 7 3 135  7  2  3  3  3  3  7  7  3 1  3  3  1  3  7  7  3  3  7  3  3  7 23  3  7
10 8 11  9 2 0   3  1 42 10 10 11 10  2 11  3 8  9  9 13  0  1  1  5 10 11 11 15  2  3  3  7
 2 0  3  1 3 1   7  3 10  2  2  3  2  3  3  3 0  1  1  5  1  3  3  1  2  3  3  7  3  7  3  3
```

Table 21: $Q_{17} = \{0,0,-1,0,0,0,0,0,-1,0,-2,0,0,1,0,1,1,1,1,0,1,2,-1,2,-1,-2,0,-2,0,0,0\}$

```
 0  3 12  15  1 19 13 31  4  7  28  31  5 23 29  95  1 35 13  47  3 51 15  63  5 39 29  15   7 55 13 31
 1  7 13  79  3 23 15 95  5 15  29  95  7 31 31 223  3 39 15 111  7 55 31 127  7  7 31  79  15 23 15 95
 4  7 44  47  5 23 45 15 12 15  60  63 13  7 61  31  5 39 45 111  7 55 13  47 13  7 61  47  15 23 29 15
 5 15 45 111  7  7 47 79 13 31  61 127 15 15 63  95  7 47 47 239 15 39 15 111 15 15 63 111 271  7 31 79
16 19 28  31 17 51 29 63 20 23  60  63 21 55 61  31 17 51 29  63 19 307 31 55 21 55 28  31  23 311 29 63
 0  3 12  15  1 19 13 31  4  7  28  31  5 23 29  95  1 35 13  47  3 51 15  63  5 39 29  15   7 55 13 31
20 23 60  63 21 55 61 31 28 31 572  61 29 23 60  63 21 55 61  47 23 311 29 63 29 23 60  63  31 55 61 31
 4  7 44  47  5 23 45 15 12 15  60  63 13  7 61  31  5 39 45 111  7 55 13  47 13  7 61  47  15 23 29 15
```

Table 22: $Q_{18} = \{0,0,0,-1,-1,0,-1,1,1,0,0,-1,0,1,1,1,-1,1,0,0,-1,1,0,1,0,0,-1,0,0,-1,0\}$

```
0 1 1  9  2  3 3 11  2  3  3 11 10 11 11 27 1  5  5  13  3  7 7 15  0 1 1  9  2  3 3 11
1 3 3  1  3 19 3  3 35  3  2  3 19  3  7  1 5  7 23   3  3  7 1  3  3 19 3  3 19 3 3
2 3 0  1  6  7 2  3  6  7  2  3 14 15 10 11 3  7  1   5  7 23 3  7  2 3 0  1  6  7 2  3
3 7 1  3  7 23 3 19  2  3  3  3  6  7  2  3 7 23  3   7 23 87 7 23  3 7 1  3  7 23 3 19
4 5 5 13  6  7 7 15  6  7  7 15 14 15 15 11 5 13 13 141  7 15 5 13  4 5 5 13  6  7 7 15
0 1 1  5  2  3 3  7  2  3  3  7  6  7  7  3 1  5  5  13  3  7 1  5  0 1 1  5  2  3 3  7
6 7 4  5 14 15 6  7 14  6  6  7 46 14 14 15 7 15  5  13 15  7 7 15  6 7 4  5 14 15 6  7
2 3 0  1  6  7 2  3  6  2  2  3 14  6  6  7 3  7  1   5  7 23 3  7  2 3 0  1  6  7 2  3
```

Table 23: $Q_{19} = \{0,0,0,0,-1,0,0,-1,1,-1,0,0,-1,0,1,1,1,0,1,-1,0,2,1,-1,1,-2,0,0,0,-1,0\}$

```
 0  1  1  17 3  7  7 23  2  3  3 19  7 39 23  55  2  3  3 19  7  15  15  7   6   7  7 23 15 47  7  39
 2  3  3  19 7 23 71 87  6  7  7 23 23 55 87 119  6  7  7 23 15   7  79 71  22  23 23  7  7 39 71 103
 1  9  9  25 7 15 15 31  3 11 11 27 15 47 31  63  3 11 11 27 15  47   7 15   7  15 15 31 47 175 15 47
 0  1  1  17 3  7  7 23  2  3  3 19  7 39 23  55  2  3  3 19  7  15  15  7   6   7  7 23 15 47  7  39
16 17 17  25 1  3  3 19 18 19 19 27  3  7  7  23 18 19 19 27  3   7   7  3  22  23 23 31  7 15  7   7
18 19 19  27 3  7  7 23 22 23 23 19  7 23 23  55 22 23 23 31  7   7  15  7 150 151 22 23 23  7  7  39
17 25 25 281 3 11 11 27 19 27 27 25  7 15 15  31 19 27 27 25  7  15   3 11  23  31 31 27 15 47  7  15
16 17 17  25 1  3  3 19 18 19 19 27  3  7  7  23 18 19 19 27  3   7   7  3  22  23 23 31  7 15  7   7
```

Table 24: $Q_{20} = \{0,0,0,0,0,0,-1,0,-1,-1,0,-1,0,0,0,2,-1,2,1,1,1,1,1,1,-1,-1,-2,-1,0,0,0\}$

```
0    1    2    3    8    9   10   11    1   17    3   19    9   25   11   27    2    3    6    7   10   11   14   15    3   19    7   23   11   27   15   31
8    9   10   11   24   25   26   27    9   25   11   27   25   57   27   59   10   11   14   15   26   27   30   31   11   27   15   31   27   59   31   27
3    7    7   71   11   15   15   79    7   23   23   87  153  131   95    7   23   23   87   15   31   31   95  263  279  279  343    7   23   23   87
11   15   15    7   27   11   31   15   15   31   31   23  112   15   31   15   31   31   23   31   15   15   31  271  287  287  279   15   31   31   23
6    7   14   15   14   15  142  143    7   23   15   31  153   14   15   14   15   30   31   30   31  158  159   15   31   31   15   31   15   30   31
14   15   30   31   30   31  158  159   15   31   31   15  316   30   31  526  145   42   30  542   30  670  158  527   15  543   31  543   31   54   30
15   47   47  111   47  111  175  239   47   15   15   79   15   47   47  111   47   15   15   79   15   79  143  207  303  271  271  335   47   15   15   79
47   15   15   47   15   47  143  175  303   47  271   15   47   15   15   47  559   47  527   15  527   15  655  143  815  303  783  271  559   47   52   15
```

Table 25: $Q_{21} = \{-3,0,-1,0,0,2,2,1,2,1,0,-2,-1,1,0,-1,-1,0,0,-1,-1,1,0,2,1,0,0,0,-2,0\}$

```
0     1     3     7     1     3    19    23     1     3     7    15     3    11     3     7     1     3    19     3     3    35    51    19     3    11     3    11    35    43    19     3
12    13    15   143     4     5     7    15    13    15   143   399     5     7    15   143     4     5     7    15     0     1     3     7     5     7    15   143     1     3     7    15
4     5     7    23     5     7    23    87     0     1     3     7     1     3     7    23     0     1     3    19    83     1     3     3     3     3    11     3    19
28    29    31    15    12    13    15     7    12    13    15   143     4     5     7    15    12    13    15     7     4     5     7     3     4     5     7    15     0     1     3     7
40    41     0     1    41    43     1     3    43   107     3    11    41    43     1     3    43   107     3    35    43   107     3    11   107   619    35    43
60    61    12    13    44    45     4     5    61    45    13    15    45    47     5     7    44    45     4     5    40    41     0     1    45    47     5     7    41    43     1     3
44    45     4     5    45    47     5     7    40    41     0     1    41    43     1     3    40    41     0     1    41    43     1     3    41    43     1     3    43   107     3    11
1084   60    28    29    60    28    29    60    44    12    13    44    45     4     5    60    44    12    13    44    45     4     5    44    45     4     5    40    41     0     1
48    16    51    19    49    17   179    51    16     0    19     3    17     1    51    19    49    17   179    51    51    19   435   179    17     1    51    19    19     3   179    51
60    28    63    31    52    20    55    23    28    12    31    15    20     4    23     7    52    20    55    23    48    16    51    19    20     4    23     7    16     0    19     3
52    20    55    23    53    21    51    19    20     4    23     7    21     5    19     3    48    16    51    19    49    17   179    51    16     0    19     3    17     1    51    19
1084   60    61    63    60    28    63    31    60    28    29    31    28    12    31    15    60    28    63    31    52    20    55    23    28    12    31    15    20     4    23     7
60    56    48    16    61    57    49    17    56    40    16     0    57    41    17     1    56    57    49    17    57    59    51    19    57    41    17     1    59    43    19     3
3132  1084   60    28  1084    60    52    20  1084    60    28    29    60    28    12    60    52    20    60    56    48    16    60    28    29     4    56    40    16     0
1084   60    52    20    60    61    53    21    60    44    20     4    61    45    21     5    60    56    48    16    56    57    49    17    56    40    16     0    57    41    17     1
7228  3132  1084   60  3132  1084    60    28  3132  1084    60    28  1084    60    28    12  3132  1084    60    28  1084    60    52    20  1084    60    28    12    60    44    20     4
```

Table 26: $Q_{22} = \{-2,0,-1,0,0,1,1,2,1,0,-1,0,1,0,-1,0,0,-1,-2,0,0,1,1,1,1,0,0,0,0,-2,0\}$

```
0     1     1     3     3     3    19     1     3     3     7     3     7    19     3     1     3     3     7     3     7     1     3     3     7     7    15     7    15     3     7
16    17    17    19    17    19    19    51     0     1     1     3     1     3    19     3     1     3     3     7     3     7     1     3     3     7    15     7     3     1     3
12    13    13    15     4     5     5     7    13    15    15    47     5     7     7    39    13    15    15    47     5     7     7    15    15    47    47   111     7    15    15    47
28    29    29    31    20    21    21    23    12    13    13    15     4     5     5     7    12    13    13    15     4     5     5     7    13    15    15    47     5     7     7    15
4     5     0     1     5     7     1     3     5     7     1     3     7    15     3     7     5     7     1     3     7    15     3     7    15   143     7    15
20    21    16    17    21    23    17    19     4     5     0     1     5     7     1     3     4     5     0     1     5     7     1     3     5     7     1     3     7    15     3     7
28    29    12    13    13     4     5     7    12    13    13    15    13     5     5     7    12    13    13    15     4     5     5     7    13    15    15    47     5     7     7    15
284   28    28    29    28    29    20    21    28    29    12    13    12    13     4     5    28    29    12    13    12    13     4     5    12    13    13    15     4     5     5     7
16     0    17     1    17     1    19     3    17     1    19     3    19     3    83    19     0     1     1     3     1     3     3     1     1     3     3     7     3     7    19     3
20    16    21    17    21    17    17    19    16     0    17     1    17     1    19     3     4     0     5     1     5     1     1     3     0     1     1     3     1     3     3     1
28    12    29    13    20     4    21     5    29    13    31    15    21     5    23     7    12    13    13    15     4     5     7    13    15    15    47     5     7     7    15
284   28    28    29    28    20    20    21    28    12    29    13    20     4    21     5    28    12    12    13    12     4     4     5    12    13    13    15     4     5     5     7
20     4    16     0    21     5    17     1    21     5    17     1    23     7    19     3     4     5     0     1     5     7     1     3     5     7    15     3     7
28    20    20    16    29    21    21    17    20     4    16     0    21     5    17     1    12     4     4     0    13     5     5     1     4     5     0     1     5     7     1     3
284   28    28    12    28    12    20     4    28    29    29    13    29    13    21     5    28    12    12    13    12     4     4     5    29    13    13    15    13     5     5     7
796   284   284   28   284    28    28    20   284    28    28    12    28    12    20     4   284    28    28    12    28    12    12     4    28    12    12    13    12     4     4     5
```

Table 27: $Q_{23} = \{-2,0,0,0,0,1,0,1,2,-1,0,-1,-1,0,-1,1,-1,0,1,-2,0,2,2,1,2,-1,0,0,0,-3,0\}$

```
0     3     1     7     3    23     7    55     3    15     7    79    15    63    79    31     8    11     9    15    11    31    15    63    11   271    15   335    31   319    15   287
8    11     9    15    11    31    15    63    11    31    15    15  1039  1087  1103  1055    24    27    25    31    27    63    31    31    27   287    11   271  1055  1343  1039  1311
48    51    49    55    51   183    55   695     0     3     1     7     3    23     7    55    56    59    57    63    59    55    63   183     8    11     9    15    11    31    15    63
56    59    57    63    59    63    567     8    11     9    15    11    31    15    63   120   123   121   127   123   127   123    51   127    55    24    27    25    31    27    63    31    31
8    11     9    15     1     7     3    23    11    79    15   207     7    31    15    15    72    75    73    79     9    15    11    31    75   335    79   463    15   287    15   271
24    27    25    31     9    15    11    31    27    95    11    79    15    63    79    31    88    91    89    95    25    31    27    15    91   351    75   335    31   319    15   287
56    59    57    63    49    59    55    51   183     8    11     9    15     1     7     3    23   120   123   121   127    57    63    59    55    72    75    73    79     9    15    11    31
120   57    56    59    57    63    59    55    24    27    25    31     9    15    11    31  2168  121   120   123   121   59   123    63    88    91    89    95    25    31    27    15
8     1     9     3    11     7    15    23    11     7    15    15    79    31   591    15    24     9    25    11    27    15    31    31    27    15    11    79    95    63    79    31
72     9    73    11    75    15    79    31    75    15    79    15  1103  1055  1615  1039    88    25    89    27    91    31    75    15  1119  1087  1103  1055
56    49    57    51    59    55    63   183     8     1     9     3    11     7    15    23   120    57    56    59    57    63    59    55    24     9    25    11    27    15    31    31
120   57   121   59   123    63   127    55    72     9    73    11    75    15    79    31   120   123   121   59   123    63    88    25    89    27    91    31    95    15
24     9    25    11     9     3    11     7    27    15    11    79    15    15    79     7    88    73    89    75    25    11    27    15    91    79    75   207    31    31    15    15
88    25    89    27    73    11    75    15    91    31    75    15    79    31   591    15   120    89    27    91    89    27    91    31    75    15    89    79    95    63    79    31
120   57    56    59    57    51    59    55    24     9    25    11     9     3    11     7  2168  121   120   123    56    59    57    63    88    73    89    75    25    11    27    15
2168  56   120    57   121   59   123    63    88    25    89    27    73    11    75    15  6264   120  2168  121   120    57   121   59   120    89   121   91    89    27    91    11
```

Table 28: $Q_{24} = \{-2,0,0,0,0,1,1,1,1,0,-1,0,0,-2,0,0,0,0,-1,1,-1,1,1,1,2,0,-1,0,0,-2,0\}$

```
0     3     1    11     1     7     3    15     1    19     3    27     3    23     7    31     4     7     5    15     5    39     7    47     5    23     7    31     7    55    15    63
4     7     5    15     5    15     7    47     0     3     1    11     1     7     3    15    12    15    13    47    13    47    15   111     4     7     5    15     5    39     7    47
8    11     9    27     3     1    11     9    27    11   155     1    19     3    27    12    15    13    31    15    27     5    23     7    31
12    15    13    31     4     7     5    15     8    11     9    27     0     3     1    11    28    31    29    15    12    15    13    47    12    15    13    31     4     7     5    15
4     7     0     3     5    23     1     7     5    23     1    19     7    55     3    23    20    23     4     7    21    55     5    39    21    55     5    23    23   119     7    55
12    15     4     7    13    31     5    15     4     7     0     3     5    23     1     7    28    31    12    15    29    63    13    47    20    23     4     7    21    55     5    39
12    15     8    11     4     7     0     3    13    31     9    27     5    23     1    19    28    31    12    15    20    23     4     7    29    23    13    31    21    55     5    23
28    13    12    15    12    15     4     7    12    15     8    11     4     7     0     3   284    29    28    31    28    31    12    15    28    31    12    15    20    23     4     7
4     1     5     3     5     3     7     7     5     3     7    11     7     7   135    15    12     5    13     7    13     7    15    15    13     7    15    15    15    23     7    31
12     5    13     7     4     7     5    15     4     1     5     3     5     3     7     7    28    13    29    15    12    15    13    47    12     5    13     7    13     7    15    15
12     9    13    11     4     1     5     3    13    11    15    27     5     3     7    11    28    13    12    15    12     5    13     7    12    15    13    11    13     7    15    15
28    13    12    15    12     5    13     7    12     9    13    11     4     1     5     3   284    29    28    31    28    13    29    15    28    13    12    15    12     5    13     7
12     5     4     1    13     7     5     3    13     7     5     3    15    23     7     7    28    21    12     5    29    23    13     7    29    23    13     7    31    55    15    23
28    13    12     5    12    15     4     7    12     5     4     1    13     7     5     3   284    29    28    13    28    31    12    15    28    21    12     5    29    23    13     7
28    13    12     9    12     5     4     1    12    15    13    11    13     7     5     3   284    29    28    13    28    21    12     5    28    31    12    15    29    23    13     7
284   12    28    13    28    13    12     5    28    13    12     9    12     5     4     1   796    28   284    29   284    29    28    13   284    29    28    13    28    21    12     5
```

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
