# Peer review of "The Quantum Entropy Cone of Hypergraphs"

_SciPost Physics, doi:SciPost Phys. 9, 067 (2020)_

## Round 1 · Referee Report · Anonymous (Referee 1) · 2020-7-28

Strengths

1- Original idea, generalizing successful techniques in a way that could prove useful to two fields (Quantum information and Quantum gravity).

2- Clear statement of past work and new results

Weaknesses

1- Leaves unproven some big, seemingly provable conjectures (such as containment of hypergraph cone in stabilizer cone for all number of parties)

Report

This paper presents a novel way of thinking about GHZ entanglement, using graph theoretic techniques. This opens new possibilities, potentially allowing the well-developed field of graph theory to help fully classify entangled states. I recommend this paper be published as is, no big revisions necessary.

It would be exciting if there were a graph theoretic way to understand all forms of entangled states. It would also be exciting if this were provably not possible.

It is my understanding that a conjecture from this paper was later proven by https://arxiv.org/abs/2002.12397. That is satisfying, because as explained in this paper under review, it seemed like this conjecture should be true!

Requested changes

Is good as is.

---

## Round 1 · Referee Report · Anonymous (Referee 2) · 2020-10-13

Strengths

  1. Clear discussion of context and review of previous related literature on the holographic entropy cone etc; section 2 sets up the required definitions related to graphs and so on.
  2. Main results are laid out as theorems and proofs, making manifest any underlying assumptions.
  3. Concluding discussion section explains additional conjectures postulated by the authors and also relates the work to higher derivative gravity and non-holographic bits.

Weaknesses

  1. While the authors link their results with previous (technical) literature, they could make a stronger case for the applications and importance of hypergraphs and their entropy cone. The results are quite technical - very accessible to those who are following the quantum information literature closely, but the overall importance of these results may be underemphasised.

Report

This is a strong paper within a field of considerable current interest to both the quantum information community and the holography (spacetime reconstruction) community. The authors present their technical results quite well - good introductory and review material, clear statements of theorems and proofs - but they could have given more discussion about the potential wider implications of these results.

---

## Editorial Decision

published